# From Abstract to Contextual: What LLMs Still Cannot Do in Mathematics

**Bowen Cao**[1,*]  **Dongdong Zhang**[2,†]  **Yixia Li**[3]  **Junpeng Liu**[1]  **Shijue Huang**[4]  **Chufan Shi**[5]
**Hongyuan Lu**[6]  **Yaokang Wu**[7]  **Guanhua Chen**[3]  **Wai Lam**[1]  **Furu Wei**[2]

[1]The Chinese University of Hong Kong  [2]Microsoft  [3]Southern University of Science and Technology
[4]The Hong Kong University of Science and Technology  [5]University of Southern California
[6]FaceMind  [7]Carnegie Mellon University
bwcao@link.cuhk.edu.hk  dozhang@microsoft.com

## Abstract

Large language models now solve many benchmark math problems at near-expert levels, yet this progress has not fully translated into reliable performance in real-world applications. We study this gap through *contextual mathematical reasoning*, where the mathematical core must be formulated from descriptive scenarios. We introduce **ContextMATH**, a benchmark that repurposes AIME and MATH-500 problems into two contextual settings: Scenario Grounding (SG), which embeds abstract problems into realistic narratives without increasing reasoning complexity, and Complexity Scaling (CS), which transforms explicit conditions into sub-problems to capture how constraints often appear in practice. Evaluating 61 proprietary and open-source models, we observe sharp drops: on average, open-source models decline by 13 and 34 points on SG and CS, while proprietary models drop by 13 and 20. Error analysis shows that errors are dominated by incorrect problem formulation, with formulation accuracy declining as original problem difficulty increases. Correct formulation emerges as a prerequisite for success, and its sufficiency improves with model scale, indicating that larger models advance in both understanding and reasoning. Nevertheless, formulation and reasoning remain two complementary bottlenecks that limit contextual mathematical problem solving. Finally, we find that fine-tuning with scenario data improves performance, whereas formulation-only training is ineffective. However, performance gaps are only partially alleviated, highlighting contextual mathematical reasoning as a central unsolved challenge for LLMs.

## 1 Introduction

Large language models (LLMs) now dominate mathematical benchmarks, scoring nearly perfectly on AIME (OpenAI, 2024; Guo et al., 2025; OpenAI, 2025) and even reaching IMO gold[1]. Yet these successes remain confined to well-defined benchmark problems, with little sign of comparable progress on the broader reasoning skills required for real-world impact (Qian et al., 2025).

This gap reflects a fundamental divide in mathematics. On one side are well-defined abstract problems, such as algebra or analytic geometry, that can be solved through established strategies and symbolic manipulation (Polya, 1945; Schoenfeld, 2014). On the other side are scenario-based problems, ranging from financial analysis to scientific research and engineering design, where the mathematical core is conveyed through concrete narrative detail. Existing benchmarks overwhelmingly target the former (Cobbe et al., 2021a; Lightman et al., 2023; He et al., 2024), leaving the latter largely unexamined. In this work, we term this underexplored domain **contextual mathematical reasoning**: the ability to formulate and solve the mathematical problem when it is embedded in narrative scenarios with indirect or layered conditions.

---

*This work was done during an internship at Microsoft Research Asia.
†Corresponding author

[1]https://x.com/alexwei_/status/1946477742855532918

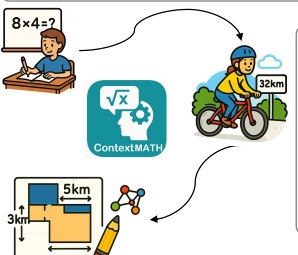

**(1) Original Problem:**
Let N denote the number of ordered triples of positive integers (a,b,c) such that $a, b, c \leq 3^6$ and $a^3 + b^3 + c^3$ is a multiple of $3^7$. Find the remainder when N is divided by 1000.

**(2) Scenario Grounding:**
…three distinct components within a sophisticated smart energy system…

…with positive values that do not exceed 729 units...

...raising their output metrics to the third power and summing these values, must yield a total that is perfectly divisible by 2,187…

…find the remainder when this number is divided by 1,000 for reporting purposes…

**(3) Complexity Scaling:**
In a simulation lab, three autonomous drones are sent out along separate axes — one moves north-south, one east-west, and one vertically. Each drone travels forward by a positive number of whole steps. These steps must remain below a certain threshold, which is known to be a power of 3. It is known that step counts as high as 2000 are not allowed, while 250 is within the allowed range. As part of a synchronization test, the system calculates a combined energy score by summing the cubes of the step counts of all three drones. Synchronization only succeeds if this total energy is also a multiple of a certain power of 3. From prior data, an energy total of 1458 fails to trigger the sync, but 8748 does trigger it. How many such ordered drone configurations satisfy both conditions? Report your answer as the remainder when this number is divided by 1000.

Figure 1: Example from ContextMATH, based on AIME 2025 Problem 15. In Scenario Grounding (SG), mathematical components are mapped to a narrative. In Complexity Scaling (CS), explicit conditions are concealed in sub-problems requiring an extra inference step. Consistent color-coding highlights correspondence between mathematical components across the three versions. LLMs remain strong on abstract benchmarks but drop accuracy on SG, with the gap widening further on CS.

To investigate this capability, we introduce **ContextMATH** (**Context**ual Reasoning Evaluation for **Math**), a benchmark designed to probe contextual mathematical reasoning systematically. Rather than seeking massive collections of real-world tasks, ContextMATH builds on standard sources—AIME 2024, AIME 2025, and MATH-500 (Lightman et al., 2023)—and instantiates each problem in two controlled narrative variants: *Scenario Grounding (SG)* embeds abstract mathematical structures into concrete narratives with real-world entities and interactions. The reasoning core remains unchanged, but the problem is situated in descriptions that naturally introduce contextual detail. *Complexity Scaling (CS)* conceals explicit conditions within sub-problems in the scenario. For example, an absolute position may be given relative to a reference point, or a numerical bound may appear as the result of a short calculation. These sub-problems yield exactly the information originally explicit, while reflecting how constraints are often expressed in real-world settings. This format also reduces reliance on surface pattern matching, since models must first interpret the scenario to recover the original conditions. Although demonstrated here for mathematics, the same approach applies to other domains and datasets, enabling systematic evaluation of contextual reasoning beyond math. Figure 1 illustrates such mappings; for example, the variables $(a, b, c)$ in the original problem correspond to system components in SG and drone movements in CS.

Our results show that both leading open-source and proprietary LLMs experience a substantial decline in contextual mathematical reasoning compared to their strong performance on original benchmarks. On average, open-source models drop 13% on SG and 34% on CS, while proprietary models fall 13% and 20%, respectively. For example, `Qwen3-32B` falls from 81.25% on AIME 2024 problems to 67.92% on SG and 57.08% on CS. A similar trend holds for proprietary systems: even `DeepSeek-R1` drops from 86.67% to 73.33% (SG) and 53.33% (CS) on AIME 2025.

To better understand these limitations, we analyze models' ability to formulate the mathematical core from narrative scenarios. We find that performance declines as problems become harder, and although larger models perform better, even `GPT-5` averages only 81.4% formulation accuracy. Furthermore, correct formulation proves to be a prerequisite: correctly solved problems show much higher formulation accuracy than average, establishing formulation as the first bottleneck. We further observe that as models scale and formulation accuracy improves, correct formulation increasingly becomes sufficient for solution correctness, indicating progress in both problem understanding

and reasoning. Yet even `GPT-5` reaches only 82.7% sufficiency, showing that reasoning remains a second major bottleneck for real-world mathematical problem solving.

We further explore whether the performance drops can be alleviated via training. Our results show that fine-tuning with scenario data effectively improves models' performance on our benchmark, yet a sizable drop on contextual variants still remains, indicating that contextual mathematical reasoning is far from solved. In contrast, directly training a dedicated formulation model proves ineffective, suggesting that formulation ability is difficult to acquire from paired supervision alone.

Our contributions can be summarized as follows:

- **New task.** We frame contextual mathematical reasoning, the ability to extract and solve the mathematical core from narrative descriptions, as a critical but underexplored capacity for LLMs.

- **New benchmark.** We propose **ContextMATH**, which instantiates problems from AIME and MATH-500 into Scenario Grounding (SG) and Complexity Scaling (CS) variants to probe mathematical formulation and reasoning under contextual challenges. Evaluations show large accuracy drops on SG and CS relative to abstract benchmarks, revealing a clear and persistent gap between abstract and contextual mathematical reasoning in current LLMs.

- **New insights.** Through quantitative and qualitative analysis, we identify formulation and reasoning as complementary bottlenecks, and provide evidence that training with scenario data improves performance, whereas training a dedicated formulation model remains ineffective. These findings suggest that contextual mathematical reasoning is learnable, but cannot be reduced to isolated skills, highlighting the need for integrated approaches to advance formulation and reasoning.

## 2 RELATED WORK

**Benchmarks on Abstract Mathematical Problems.** Much of the progress in evaluating LLMs' mathematical ability has been driven by benchmarks centered on abstract, well-specified problems. GSM8K (Cobbe et al., 2021b) covers grade school arithmetic word problems, while MATH (Hendrycks et al., 2021b) spans high-school curricula, and AIME, AMC23, and Olympiad-Bench (He et al., 2024) emphasize challenging competition-style questions. There are also benchmarks that target formal theorem proving, such as PutnamBench (Tsoukalas et al., 2024), Lean-Dojo (Yang et al., 2023), and MiniF2F (Zheng et al., 2021). While some existing problems contain simple narratives such as "Jack had 8 pens and Mary had 5 pens...," (Patel et al., 2021) these contexts are shallow and limited in scope. As such, current benchmarks primarily capture performance on clearly formulated abstract problems, leaving open the question of how well models can reason about mathematics when it is embedded in richer contexts.

**From Abstract Problems to Scenarios.** Across domains, a growing line of work has begun to examine model reasoning in realistic settings. For example, WebArena (Zhou et al., 2024) evaluates agents in interactive web environments, while SWE-bench (Jimenez et al., 2024) and BaxBench (Vero et al., 2025) focus on real-world software engineering tasks. Our work falls within this broader scope, but focuses specifically on mathematical reasoning in scenarios.

## 3 PROBING LLMS' ABILITY TO SOLVE MATH IN SCENARIOS

### 3.1 OVERVIEW

A central question of this work is whether LLMs possess *contextual mathematical reasoning*—the ability to formulate and solve the mathematical core when problems are embedded in descriptive scenarios. This capacity is crucial for applying mathematics in practical domains, where tasks rarely appear as ready-made equations. Collecting large numbers of real-world instances, however, is costly and difficult to scale. We therefore leverage established benchmarks with guaranteed correctness and systematically transform them into contextual variants. Concretely, we build **ContextMATH**, which instantiates each benchmark problem in two forms: *Scenario Grounding*, preserving the original structure but embedding it in narrative context; and *Complexity Scaling*, encoding explicit conditions into simple sub-problems. We then evaluate a diverse set of 46 open-source models and 15 proprietary models, including `GPT-5` and `DeepSeek-R1`.

| Model | AIME 2024 (%) | | | | AIME 2025 (%) | | |
|---|---|---|---|---|---|---|---|
| | **Ori** | **SG** | **SG Avg@3** | **CS** | **Ori** | **SG** | **CS** |
| Copilot | 30.0 | 30.0 ( -0% ) | 28.9 ( -4% ) | 23.3 ( -22% ) | 33.3 | 20.0 ( -40% ) | 16.7 ( -50% ) |
| gpt-4o-mini | 6.7 | 3.3 ( -50% ) | 6.7 ( -0% ) | 3.3 ( -50% ) | 10.0 | 6.7 ( -33% ) | 0.0 ( -100% ) |
| o1-mini | 60.0 | 53.3 ( -11% ) | 53.3 ( -11% ) | 40.0 ( -33% ) | 40.0 | 33.3 ( -17% ) | 23.3 ( -42% ) |
| gpt-4o | 16.7 | 13.3 ( -20% ) | 11.1 ( -33% ) | 3.3 ( -80% ) | 10.0 | 6.7 ( -33% ) | 0.0 ( -100% ) |
| gpt-4.1-mini | 46.7 | 26.7 ( -43% ) | 34.4 ( -26% ) | 30.0 ( -36% ) | 53.3 | 30.0 ( -44% ) | 16.7 ( -69% ) |
| gpt-4.1-nano | 26.7 | 23.3 ( -13% ) | 18.9 ( -29% ) | 6.7 ( -75% ) | 33.3 | 20.0 ( -40% ) | 13.3 ( -60% ) |
| R1 | **93.3** | 70.0 ( -25% ) | 70.0 ( -25% ) | 66.7 ( -29% ) | 86.7 | 73.3 ( -15% ) | 53.3 ( -38% ) |
| Doubao-1.5 | 90.0 | 70.0 ( -22% ) | 66.7 ( -26% ) | 56.7 ( -37% ) | 76.7 | 53.3 ( -30% ) | 43.3 ( -43% ) |
| Qwen-max | 23.3 | 16.7 ( -29% ) | 14.4 ( -38% ) | 6.7 ( -71% ) | 13.3 | 10.0 ( -25% ) | 0.0 ( -100% ) |
| QwQ-plus | 86.7 | 56.7 ( -35% ) | 60.0 ( -31% ) | 46.7 ( -46% ) | 73.3 | 53.3 ( -27% ) | 43.3 ( -41% ) |
| Grok3 | 43.3 | 23.3 ( -46% ) | 22.2 ( -49% ) | 23.3 ( -46% ) | 26.7 | 13.3 ( -50% ) | 20.0 ( -25% ) |
| Gemini 2.5 Flash | 70.0 | 63.3 ( -10% ) | 61.1 ( -13% ) | 53.3 ( -24% ) | 70.0 | 43.3 ( -38% ) | 30.0 ( -57% ) |
| Gemini 2.5 Pro | 83.3 | 73.3 ( -12% ) | 68.9 ( -17% ) | 76.7 ( -8% ) | 83.3 | 56.7 ( -32% ) | 50.0 ( -40% ) |
| o3 | 83.3 | 70.0 ( -16% ) | 73.3 ( -12% ) | 66.7 ( -20% ) | 76.7 | 70.0 ( -9% ) | 60.0 ( -22% ) |
| gpt-5 | 90.0 | **83.3** ( -7% ) | **82.2** ( -9% ) | **80.0** ( -11% ) | **90.0** | **80.0** ( -11% ) | **66.7** ( -26% ) |

Table 1: Accuracy of proprietary models on ContextMATH. Best and second-best results per column are in **bold** and underlined, respectively. Parentheses indicate the relative performance change from Ori, with larger drops highlighted in a deeper shade of red. To verify that our findings are consistent and not an artifact of specific scenarios, we additionally generated and annotated two AIME2024-SG sets, reporting the average accuracy across all three in the `SG Avg@3` column.

## 3.2 BENCHMARK CONSTRUCTION

Our benchmark is constructed from AIME 2024, AIME 2025, and MATH-500[2]. Our design follows two principles: preserving mathematical equivalence and embedding problems in realistic narrative contexts. To realize this, we employ a series of structured prompts that guide an LLM (`o1-mini`) through iterative scenario generation, self-verification, and revision. Human experts then review and refine each item to guarantee mathematical equivalence, clarity and conciseness[3].

**Scenario Grounding (SG): Evaluating the Application of Math in Context.** The primary goal of scenario grounding is to assess whether a model can apply its mathematical knowledge when problems are presented in a narrative context, without increasing the core reasoning difficulty. To achieve this, our generation process is guided by a multi-step prompt that instructs the model to first explicitly map all abstract mathematical elements to specific real-world objects (e.g., mapping *"a variable x"* to *"the initial number of barrels of oil"*). The prompt then directs the model to define the rules of interaction between these objects based on the attributes and relationship in the problem. This structured approach ensures that all mathematical components of the original problem are preserved in the final scenario (see Appendix A.1 for the full prompt).

**Complexity Scaling (CS): Encoding Implicit Constraints through Sub-Problems.** In many real-world settings, quantitative conditions must be inferred from indirect descriptions, such as simple calculations, relative positions, or everyday facts. To reflect this natural form of complexity, CS problems transform some direct conditions into the outputs of simple, self-contained sub-problems. For example, instead of stating there are *"25 indicator lights"*, the problem may specify that *"the total number of unique pairs of indicator lights is exactly 300"*. Our generation prompt provides principled strategies for such transformations, including encoding values as the solutions to number theory or combinatorics problems, replacing explicit functions or constants with variables to be determined from given data points, and rephrasing geometric relationships in terms of physical or structural descriptions. This design also provides a more faithful and robust test of a model's rea-

---

[2]We select only problems with difficulty level $\geq 3$ from MATH-500, since easier items are less suitable for embedding into meaningful scenarios. For simplicity, we continue to refer to this filtered subset as MATH-500 throughout the paper. We do not construct the CS set for MATH-500 because some remaining items are too trivial to transform further, e.g., 'Evaluate $(1 + 2i)(6 - 3i)$.'.

[3]The average lengths of SG and CS problems are 133 and 176 words, respectively, both well within the processing capabilities of current LLMs.

| Model | AIME 2024 (%) | | | AIME 2025 (%) | | | Math-500 (%) | |
|---|---|---|---|---|---|---|---|---|
| | **Ori** | **SG** | **CS** | **Ori** | **SG** | **CS** | **Ori** | **SG** |
| *≤4B* | | | | | | | | |
| Qwen3-0.6B | 7.9 | 6.2 ( -21% ) | 0.4 ( -95% ) | 15.4 | 5.4 ( -65% ) | 2.5 ( -84% ) | 62.4 | 42.6 ( -32% ) |
| Qwen2.5-Math-1.5B | 11.5 | 2.5 ( -79% ) | 0.4 ( -96% ) | 4.6 | 0.8 ( -82% ) | 0.0 ( -100% ) | 32.5 | 25.8 ( -21% ) |
| ↪R1-Distil-Qwen-1.5B | 28.3 | 20.0 ( -29% ) | 7.1 ( -75% ) | 19.6 | 9.6 ( -51% ) | 5.8 ( -70% ) | 74.0 | 57.6 ( -22% ) |
| ↪DeepScaleR-1.5B-Preview | 41.5 | 23.3 ( -44% ) | 7.5 ( -82% ) | 30.4 | 15.4 ( -49% ) | 7.5 ( -75% ) | 80.1 | 64.2 ( -20% ) |
| ↪OpenMath-Nemotron-1.5B | 62.5 | 34.2 ( -45% ) | 14.6 ( -77% ) | 50.4 | 21.2 ( -58% ) | 11.2 ( -78% ) | 87.3 | 70.1 ( -20% ) |
| ↪DeepMath-Omn-1.5B | 62.9 | 33.8 ( -46% ) | 13.8 ( -78% ) | 55.8 | 20.8 ( -63% ) | 12.5 ( -78% ) | 87.1 | 73.0 ( -16% ) |
| Qwen3-1.5B | 46.2 | 29.6 ( -36% ) | 12.5 ( -73% ) | 34.6 | 24.6 ( -29% ) | 11.2 ( -67% ) | 84.1 | 67.4 ( -20% ) |
| Qwen3-4B | **70.4** | 52.5 ( -25% ) | **34.6** ( -51% ) | **64.2** | 39.6 ( -38% ) | **33.8** ( -47% ) | **90.9** | 78.8 ( -13% ) |
| *7B/8B* | | | | | | | | |
| Qwen2.5-Math-7B | 10.8 | 6.7 ( -38% ) | 1.5 ( -85% ) | 5.0 | 3.3 ( -33% ) | 0.8 ( -83% ) | 44.8 | 36.7 ( -18% ) |
| ↪OpenMath-Nemotron-7B | 72.9 | 52.1 ( -29% ) | 30.0 ( -59% ) | 60.0 | 40.4 ( -33% ) | 29.2 ( -51% ) | 90.5 | 78.5 ( -13% ) |
| ↪R1-Distil-Qwen-7B | 48.8 | 40.0 ( -18% ) | 23.3 ( -52% ) | 41.5 | 22.5 ( -46% ) | 15.0 ( -64% ) | 87.4 | 73.9 ( -15% ) |
| ↪AceMath-RL-Nemotron-7B | 69.2 | 48.8 ( -30% ) | 32.9 ( -52% ) | 54.2 | 26.7 ( -51% ) | 22.5 ( -58% ) | 89.9 | 79.0 ( -12% ) |
| Qwen3-8B | 73.8 | **61.5** ( -16% ) | **42.9** ( -42% ) | 64.6 | **48.3** ( -25% ) | 35.8 ( -45% ) | **91.0** | 81.0 ( -11% ) |
| ↪R1-0528-Qwen3-8B | **75.0** | 55.0 ( -27% ) | 39.6 ( -47% ) | 65.8 | **48.3** ( -27% ) | 32.9 ( -50% ) | 90.7 | 77.2 ( -15% ) |
| ↪AReaL-boba-2-8B | 74.2 | 58.3 ( -21% ) | 41.5 ( -44% ) | **67.9** | 47.9 ( -29% ) | **37.1** ( -45% ) | **91.5** | 82.0 ( -11% ) |
| *14B* | | | | | | | | |
| Qwen2.5-14B | 6.2 | 3.8 ( -40% ) | 1.5 ( -73% ) | 3.3 | 1.5 ( -50% ) | 0.0 ( -100% ) | 48.5 | 34.9 ( -28% ) |
| ↪R1-Distil-Qwen-14B | 67.5 | 47.1 ( -30% ) | 35.4 ( -48% ) | 50.8 | 26.2 ( -48% ) | 25.8 ( -49% ) | 89.5 | 76.9 ( -14% ) |
| ↪OpenMath-Nemotron-14B | 73.8 | 51.5 ( -30% ) | 42.1 ( -43% ) | 63.8 | 42.5 ( -33% ) | 29.2 ( -54% ) | 90.8 | 80.8 ( -11% ) |
| Qwen3-14B | 80.0 | 64.6 ( -19% ) | 50.8 ( -36% ) | 72.9 | 49.2 ( -33% ) | **42.1** ( -42% ) | 92.6 | 81.9 ( -12% ) |
| ↪AReaL-boba-2-14B | **82.9** | 65.8 ( -21% ) | **53.8** ( -35% ) | **73.3** | 52.1 ( -29% ) | 39.2 ( -47% ) | 91.9 | 82.9 ( -10% ) |
| Phi-4-reasoning-plus | 80.4 | 60.4 ( -25% ) | 52.9 ( -34% ) | 71.5 | **55.4** ( -22% ) | 39.6 ( -45% ) | 92.6 | 83.1 ( -10% ) |
| *≥32B* | | | | | | | | |
| Qwen2.5-32B | 11.2 | 6.7 ( -41% ) | 3.3 ( -70% ) | 3.8 | 2.9 ( -22% ) | 0.0 ( -100% ) | 45.2 | 37.8 ( -16% ) |
| ↪OpenMath-Nemotron-32B | 57.1 | 42.5 ( -26% ) | 27.9 ( -51% ) | 52.1 | 34.6 ( -34% ) | 27.5 ( -47% ) | 75.8 | 61.8 ( -18% ) |
| ↪R1-Distil-Qwen-32B | 69.6 | 52.5 ( -25% ) | 39.2 ( -44% ) | 56.2 | 39.6 ( -30% ) | 30.0 ( -47% ) | 89.4 | 78.9 ( -12% ) |
| Qwen3-32B | 81.2 | **67.9** ( -16% ) | 57.1 ( -30% ) | 70.0 | 54.4 ( -22% ) | **45.0** ( -36% ) | 92.1 | 82.7 ( -10% ) |
| ↪AReaL-boba-2-32B | **81.5** | 65.4 ( -20% ) | **58.3** ( -29% ) | **77.1** | 55.0 ( -29% ) | 43.8 ( -43% ) | 92.3 | 82.9 ( -10% ) |
| QwQ-32B | 80.4 | 58.3 ( -27% ) | 53.3 ( -34% ) | 66.2 | 53.3 ( -20% ) | 39.2 ( -41% ) | 92.5 | 82.9 ( -10% ) |
| R1-Distill-Llama-70B | 65.4 | 48.8 ( -25% ) | 38.8 ( -41% ) | 50.0 | 38.8 ( -22% ) | 29.2 ( -42% ) | 89.8 | 77.3 ( -14% ) |

Table 2: Accuracy of open-source models on ContextMATH. Models are grouped by parameter scale, and best and second-best results within each group are shown in **bold** and underlined, respectively. Parentheses indicate the relative performance change from Ori, with larger drops highlighted by a deeper shade of red. Arrows (↪) denote variants obtained from the model listed directly above. Full results are provided in Appendix A.3.

soning ability by reducing reliance on superficial pattern matching, a behavior widely documented in prior studies showing the limited generalization of LLMs (Cao et al., 2024; Sun et al., 2025).

**Quality Control.** We place high importance on data quality. Our approach involves two key measures: first, we have optimized the instructions and generation pipeline through extensive pilot studies; second, our procedure involves three experts, all with advanced degrees in Computer Science and backgrounds in competitive mathematics, to fundamentally ensure accuracy. They conduct independent reviews based on the following criteria:

- Assessing the scenario for narrative plausibility and clarity, ensuring it introduces no unnecessary complexity or ambiguity.
- Independently formulating an abstract math problem from the scenario to verify its solvability and mathematical equivalence to the original.
- Testing the scenario on Gemini and GPT-5; if these models fail, the reviewer diagnoses whether the failure originates from an ambiguity in the problem description.

A scenario is accepted only after passing all checks. Any identified issue flags the scenario for revision. In cases of disagreement, the reviewer with the strongest mathematical background leads a discussion to reach a consensus, which may trigger a regeneration or revision cycle.

## 3.3 EVALUATION SETUP

**Models.** We evaluate 46 advanced open-source models (including base, SFT, and RL-tuned) and 15 frontier proprietary models[4]. Details are provided in Appendix A.2.

**Metrics.** We use accuracy as the performance metric. For open-source models, we sample 16 solutions per problem using their recommended generation configurations and report the average accuracy. For proprietary models, which are accessed via APIs or client interfaces that often have rate limits or higher costs, we report the accuracy of a single-pass evaluation.

## 3.4 RESULTS

Our main experimental results, presented in Table 1 and 2, reveal three clear insights:

**Contextual Complexity as a Universal Bottleneck.** Both open-source and proprietary models show consistent and severe performance drops on contextual scenarios. For instance, `DeepSeek-R1-0528-Qwen3-8B` falls from 75.0% on AIME 2024 to 39.6% on its CS variant, and `QwQ-plus` from 86.7% to 46.7%. Even `GPT-5`, with an estimated 1.8T parameters, drops 26% on AIME 2025-CS. These findings make clear that contextual mathematical reasoning remains a fundamental and unresolved challenge, underscoring that progress on abstract math does not fully translate into robust performance in contextual settings.

**Scale Mitigates but Does Not Solve the Problem.** In open-source evaluations, larger models retain more robustness, but scaling does not eliminate contextual failure. In the `OpenMath-Nemotron` series on AIME 2024-CS, the 1.5B model drops 77%, compared to 43% and 51% for 14B and 32B. Similar trends hold for `R1-Distil-Qwen2.5` and `Qwen3` families. Larger models are more resilient, but interpreting complex narratives remains a core bottleneck.

**Initial SFT Improves Robustness, but Further Specialization Does Not.** A first stage of supervised fine-tuning consistently improves both original and contextual accuracy. For example, `Qwen2.5-Math-7B→R1-Distil-Qwen-7B` not only boosts AIME accuracy but also reduces contextual drop. However, further SFT or RL for advanced reasoning does not improve contextual tasks. Models such as `R1-Distil-Qwen-7B→AceMath-RL-Nemotron-7B` and `Qwen3-8B→DeepSeek-R1-0528-Qwen3-8B` improve on AIME 2024/25 but not on SG or CS, often with even larger drops. This suggests current post-training can over-specialize to canonical formats, reinforcing pattern recognition rather than contextual reasoning.

# 4 ANALYSIS OF FAILURE MODES: THE FORMULATION BOTTLENECK

The performance degradation observed in Section 3 motivates an analysis of the underlying failure modes. Our central hypothesis is that these failures do not primarily stem from flawed computational reasoning, but from the incorrect interpretation of the problem's mathematical structure from its narrative context. This section validates this hypothesis through a combination of qualitative analysis and quantitative experiments that isolate contextual mathematical problem formulation skill.

## 4.1 QUALITATIVE ANALYSIS: INCORRECT MATHEMATICAL INTERPRETATION

To examine how such failures occur, we ask `GPT-5` to categorize the errors made by `DeepSeek R1`, `Gemini 2.5 Pro`, and `Qwen3-32B` on the SG and CS sets of AIME 2024 and 2025, restricted to cases where the same models solve the original version correctly.

---

[4]Since AIME 2024 and AIME 2025 results already established a performance trend for proprietary models, and given the high API costs of Math-500, we excluded proprietary models from this part of the evaluation.

| Model | Formulation Accuracy (%) | | | | Formulation Necessity (%) | | | | Formulation Sufficiency (%) | | | |
|---|---|---|---|---|---|---|---|---|---|---|---|---|
| | MATH | AIME24 | AIME25 | Avg. | MATH | AIME24 | AIME25 | Avg. | MATH | AIME24 | AIME25 | Avg. |
| R1-Distill-Qwen-1.5B | 62.3 | 48.8 | 40.4 | 48.1 | 70.6 | 52.6 | 58.3 | 58.5 | 65.0 | 13.8 | 12.5 | 23.5 |
| R1-Distill-Qwen-7B | 75.7 | 57.9 | 47.1 | 57.1 | 82.9 | 67.0 | 60.8 | 67.7 | 81.1 | 38.7 | 24.5 | 41.5 |
| R1-Distill-Qwen-14B | 85.7 | 68.8 | 51.7 | 65.3 | 90.1 | 76.2 | 79.8 | 80.4 | 80.8 | 47.9 | 37.0 | 50.1 |
| R1-Distill-Qwen-32B | 87.3 | 71.7 | 59.6 | 70.0 | 91.8 | 78.5 | 73.9 | 79.3 | 83.5 | 48.8 | 42.7 | 53.3 |
| Qwen3-0.6B | 57.4 | 37.9 | 40.4 | 42.8 | 69.2 | 76.4 | 29.2 | 56.1 | 52.2 | 6.0 | 1.7 | 13.5 |
| Qwen3-1.7B | 82.5 | 57.1 | 56.7 | 62.0 | 90.1 | 79.1 | 75.8 | 80.0 | 73.8 | 28.6 | 20.8 | 34.5 |
| Qwen3-4B | 84.8 | 64.6 | 47.1 | 61.6 | 90.3 | 85.5 | 67.4 | 79.2 | 83.6 | 57.1 | 54.2 | 61.3 |
| Qwen3-8B | 86.3 | 77.9 | 63.3 | 73.8 | 91.3 | 86.7 | 77.2 | 83.8 | 86.3 | 57.7 | 51.0 | 60.7 |
| Qwen3-14B | 87.1 | 72.9 | 64.6 | 72.4 | 90.4 | 78.5 | 88.4 | 84.8 | 85.2 | 61.0 | 62.2 | 66.3 |
| Qwen3-32B | 89.3 | 77.1 | 65.8 | 75.0 | 92.0 | 77.2 | 81.4 | 81.9 | 85.0 | 63.0 | 56.6 | 64.9 |
| gpt5 | 90.5 | 85.0 | 73.3 | 81.4 | 94.5 | 87.6 | 79.2 | 85.6 | 86.9 | 84.2 | 79.2 | 82.7 |

Table 3: Problem formulation performance across different models. Formulation Accuracy denotes how often a model correctly translates a scenario into its formulation. Formulation Necessity quantifies the extent to which correct formulations are required for correct reasoning, while Formulation Sufficiency evaluates whether correct formulations reliably lead to correct reasoning. The AIME24 and AIME25 columns show the average of their SG and CS sets, while the 'Avg.' column is the mean over all sets. Darker cell colors indicate higher values.

Errors are grouped into four categories: formulation (incorrect mapping from narrative to math), calculation, logic, and other (e.g., truncation, repetition). Figure 2 shows that formulation errors dominate, accounting for roughly 80% of failures across all three models and far exceeding any other type. A typical case is given in Appendix A.4, where `DeepSeek R1` fails to recognize that "*the time for a gear to complete one rotation is adjustable, but it must not exceed six rotations per minute*" implies the inequality $x \geq 10$, with $x$ denoting seconds per rotation. This consistent pattern confirms that formulation is the primary weakness in contextual reasoning. This presents an interesting contrast with Cheng et al. (2025): while their findings suggest that abstract formulation is not the primary bottleneck on simpler benchmarks like GSM8K (Cobbe et al., 2021a), our work demonstrates that formulation becomes a critical bottleneck once the mathematical core is embedded in complex, contextual narratives.

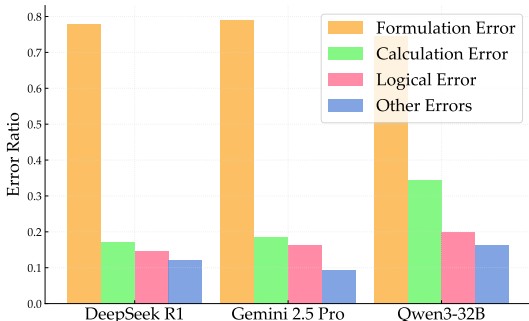

Figure 2: Distribution of error types in failure cases on AIME 2024/2025 SG and CS problems, where ratios indicate the proportion of cases exhibiting each error type.

## 4.2 QUANTITATIVE VALIDATION: EVALUATING PROBLEM FORMULATION FROM CONTEXT

In this section, we evaluate a model's ability to abstract a mathematical formulation from a given scenario. For each problem in the SG and CS test sets, the model is prompted to output only the core formulation, stripped of all narrative details. In addition to reporting formulation accuracy, we analyze the necessity and sufficiency of correct formulations for correct reasoning, highlighting the fundamental relationship between them.

**Evaluation Setups.** We carefully design the instructions and examples to prompt `o1-mini` as an automated judger, which determines whether the model's output is mathematically equivalent to the original problem.[5] For each problem, we evaluate 16 samples using the same seed set as in our benchmark evaluation (Section 3) and report three complementary metrics. The first is

---

[5]We manually annotated the outputs of `Qwen3-14B` and `Qwen3-32B` and found that the judger's assessments align with human judgments in over 90% of cases.

Formulation Accuracy , defined as the mean of the judger's assessments across samples. The second is Formulation Necessity, which measures how often a correct problem formulation is required for a correct solution. Formally, let $F$ denote whether the formulation is correct and $R$ whether the reasoning produces the correct answer. Necessity is defined as the conditional probability

$$\text{Formulation Necessity} = P(F = \text{True} \mid R = \text{True}). \tag{1}$$

This quantifies the consistency of the relation $R \Rightarrow F$. The third is Formulation Sufficiency, which evaluates whether a correct formulation reliably leads to a correct solution. It is given by

$$\text{Formulation Sufficiency} = P(R = \text{True} \mid F = \text{True}). \tag{2}$$

This corresponds to the coverage of the relation $F \Rightarrow R$. Together, these three metrics offer a coherent perspective: accuracy captures overall performance, while necessity and sufficiency jointly characterize the directional dependency between formulation and reasoning.

**Results.** To present an overview of model behavior, we report aggregated formulation metrics in Table 3, where the AIME24 and AIME25 scores are averaged over their SG and CS sets for simplicity, while the full breakdown is provided in Appendix A.5. To further analyze the relationship between formulation and overall task performance, we also combine these metrics with the reasoning accuracies from Section 3 and visualize the correlations in Figure 3. Our findings can be summarized into three key insights:

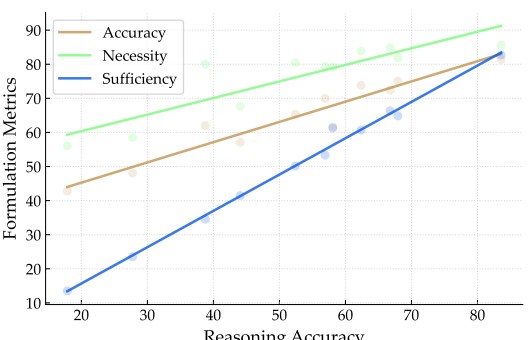

Figure 3: Relationship between reasoning accuracy and formulation metrics. Each point represents a model, with the x-axis showing its average reasoning accuracy across all subsets and the y-axis showing the corresponding values of formulation accuracy (orange), necessity (green), and sufficiency (blue). The fitted lines indicate the overall trends.

**(i) Formulation accuracy declines with problem difficulty.** Table 3 shows that formulation accuracy (orange) decreases steadily from MATH-500 to AIME24 to AIME25 (e.g., Qwen3-4B: $84.8\% \rightarrow 64.6\% \rightarrow 47.1\%$), consistent with the increasing difficulty of the underlying problems. This trend shows that problems with a harder mathematical core remain more difficult for models to formulate correctly. Scaling improves performance (e.g., Qwen3-0.6B averages 42.8% vs. 75.0% for Qwen3-32B), but even GPT-5 remains below 75% on AIME25, underscoring that the formulation challenge grows with problem difficulty and cannot be resolved by scale alone.

**(ii) Correct formulation is highly necessary for correct reasoning.** On average, necessity (green) is consistently above accuracy across all models. For example, Qwen3-4B achieves 61.6% accuracy vs. 79.2% necessity, and GPT-5 81.4% vs. 85.6%. Figure 3 confirms this pattern: necessity closely tracks reasoning accuracy but is consistently higher, indicating that correct reasoning greatly depends on having a correct formulation.

**(iii) Beyond formulation, reasoning ability still limits success.** Sufficiency (blue) improves with scale but lags behind both accuracy and necessity. In Table 3, Qwen3-8B records 73.8% accuracy and 83.8% necessity but only 60.7% sufficiency, while GPT-5 achieves 82.7% sufficiency. Figure 3 further highlights this gap: sufficiency rises with reasoning performance but consistently trails necessity, indicating that while correct formulation is indispensable, reliably completing the reasoning process remains challenging even for the strongest models, thus forming a second bottleneck.

## 5 IMPROVING CONTEXTUAL MATHEMATICAL REASONING

Our analysis in Section 4 reveals two bottlenecks: (i) formulating the mathematical core from a scenario, and (ii) carrying out the subsequent reasoning. In this section, we explore training strategies addressing these challenges, through both end-to-end fine-tuning and dedicated formulation models.

| Model | AIME 2024 (%) | | | AIME 2025 (%) | | | Math (%) | | Math-P (%) | | AMC23 | Average |
|---|---|---|---|---|---|---|---|---|---|---|---|---|
| | Ori | SG | CS | Ori | SG | CS | Ori | SG | Simple | Hard | | |
| Qwen3-4B-Base | 9.8 | 6.2 | 3.3 | 8.1 | 4.0 | 0.8 | 51.7 | 40.3 | 53.5 | 34.6 | 43.4 | 23.3 (+ 0.0%) |
| + SFT$_{Ori.}$ | 32.5 | 27.3 | 14.2 | 27.9 | 18.5 | 11.5 | 80.5 | 65.5 | 81.2 | 66.3 | 75.6 | 45.6 (+22.3%) |
| + SFT$_{Syn.}$ | 34.2 | **32.1** | 17.3 | **31.7** | 20.2 | 11.9 | 78.8 | 70.8 | 80.0 | 65.1 | 74.7 | 47.0 (+23.7%) |
| + SFT$_{Mix}$ | **36.9** | 31.7 | **19.2** | 30.2 | **22.1** | **12.7** | **81.4** | **72.3** | **82.7** | **69.0** | **78.1** | **48.8**(+25.5%) |
| Qwen3-8B-Base | 13.3 | 7.5 | 2.9 | 10.2 | 6.0 | 1.2 | 58.5 | 46.2 | 61.1 | 38.9 | 54.4 | 27.3 (+ 0.0%) |
| + SFT$_{Ori.}$ | 44.4 | 35.4 | 20.0 | 32.7 | 21.2 | 15.0 | 85.7 | 74.0 | **86.9** | 73.6 | 83.9 | 52.1 (+24.8%) |
| + SFT$_{Syn.}$ | **47.7** | **44.8** | **30.6** | **37.5** | 25.8 | **20.6** | 84.4 | 76.4 | 85.6 | 72.8 | 83.3 | 55.4 (+27.9%) |
| + SFT$_{Mix}$ | 46.2 | 42.4 | 29.5 | 35.9 | **26.8** | 20.5 | **85.9** | **76.7** | **86.9** | **74.4** | **86.6** | **55.6**(+28.3%) |
| Qwen3-14B-Base | 14.8 | 11.0 | 4.8 | 10.2 | 7.1 | 1.2 | 60.8 | 50.2 | 63.5 | 43.9 | 55.9 | 29.4 (+ 0.0%) |
| + SFT$_{Ori.}$ | 50.4 | 39.0 | 25.2 | 41.7 | 25.2 | 20.4 | 85.9 | 73.4 | 85.5 | 75.0 | 89.1 | 55.5 (+26.1%) |
| + SFT$_{Syn.}$ | **58.3** | 46.4 | **38.8** | **50.0** | 30.3 | 23.9 | 85.5 | 77.5 | **88.7** | **76.3** | 88.9 | 60.4 (+31.0%) |
| + SFT$_{Mix}$ | 56.5 | **52.5** | **38.8** | 47.2 | **34.6** | **26.5** | **86.7** | **77.8** | 88.2 | 76.1 | **89.8** | **61.3**(+31.9%) |

Table 4: Performance of `Qwen3-Base` models after supervised fine-tuning. Each base model is compared with variants trained on original problems (SFT$_{Ori}$), synthetic scenario problems (SFT$_{Syn}$), or their mixture (SFT$_{Mix}$). The best result within each block is highlighted in **bold**.

## 5.1 END-TO-END TRAINING

**Data.** (i) **Original Data**: We use DeepMath-103K (He et al., 2025), a large-scale dataset of challenging mathematical problems spanning Algebra, Calculus, and Geometry. Each problem is paired with three solutions generated by `DeepSeek R1`; we randomly sample one per problem as the reference solution. (ii) **Synthetic Scenarios**: We generate contextual scenarios for DeepMath problems following the procedure in Section 3. To enable scalable data generation without manual verification, we use `Qwen3-32B` to solve the generated scenarios and retain only those whose final answers match the original problems, ensuring a high likelihood of equivalence. This yields $50k$ validated contextual problems with model-provided reference solutions. For fairness, we also restrict the original dataset to the corresponding $50k$ subset when training.

**Experimental Setup.** We fine-tune the `Qwen3-Base` series under three training regimes for controlled comparison: (1) original data only (+SFT$_{Ori}$, $50k$), (2) synthetic scenario data only (+SFT$_{Syn}$, $50k$), and (3) a balanced mixture of both (+SFT$_{Mix}$, $100k$). All settings are trained with the same number of steps for a fair comparison. In addition to our own benchmark, we further evaluate models on AMC23 and Math-Perturb (Huang et al., 2025) to assess whether the improvements generalize beyond our benchmark. AMC23 measures abstract mathematical reasoning independent of contextualization. Math-Perturb tests generalization under data shifts, with two variants derived from level-5 MATH problems (Hendrycks et al., 2021a): *Simple*, which alters only non-critical surface parameters (e.g., numbers), and *Hard*, which modifies the underlying mathematical core. Implementation details are provided in Appendix A.6.

**Results** Table 4 shows that SFT markedly boosts contextual reasoning, e.g., `Qwen3-14B-Base` solves only 11.0% of AIME 2024-SG problems, but rises to 52.5% with SFT$_{Mix}$. Comparing regimes, scenario supervision (SFT$_{Syn.}$) is more effective than original problems (SFT$_{Ori.}$) on contextual problems, showing that models must see narratives explicitly to acquire this skill. The mixture regime (SFT$_{Mix}$) provides the best overall balance—achieving the strongest averages across all sizes—suggesting complementarity between abstract and scenario data. Beyond our benchmark, models also improve on AMC23 and Math-Perturb, indicating that targeted scenario training does not degrade abstract reasoning and even enhances robustness to distribution shifts. Larger models benefit more from scenario supervision, but even with SFT$_{Mix}$ they solve under 40% of AIME 2024/25-CS problems, highlighting both clear progress and substantial remaining headroom.

## 5.2 TRAINING A DEDICATED FORMULATION MODEL

With formulation identified as a bottleneck, a natural question is whether a model trained solely for formulation, combined with an existing solver, can improve contextual reasoning. This experiment offers a preliminary test of that idea.

**Data.** We use scenario–original pairs from Section 5.1, where each training instance maps a contextual scenario to its equivalent abstract problem.

**Experimental Setup.** We fine-tune `Qwen3-8B` and `Qwen3-14B` as *formulation models*, and also employ them as *reasoning models*. We compare three settings: (i) **w/o formulation**: the reasoning model solves the scenario directly; (ii) **w/ formulation, training-free**: the formulation model is used without fine-tuning, and its output is solved by the reasoning model; (iii) **w/ formulation, trained**: the formulation model is fine-tuned on scenario–original pairs before handing outputs to the reasoning model. At evaluation, we sample one formulation per scenario and report the average accuracy of 16 reasoning outputs. Implementation details are provided in Appendix A.6.

**Results.** As shown in Table 5, the best performance comes from solving scenarios directly (**w/o formulation**). Adding an untuned formulation stage leads to a slight drop, indicating that the pipeline introduces extra errors that propagate to reasoning. With a tuned formulation model, performance collapses further. Since training is stable and shows no signs of overfitting, we conclude that formulation is difficult to learn effectively from scenario–original pairs alone, and we leave the exploration of other approaches to future work.

| Reasoning Model | Formulation Model (Qwen3-8/14B) | | | | |
|---|---|---|---|---|---|
| | w/o | Untuned | | Tuned | |
| | | 8B | 14B | 8B | 14B |
| Qwen3-8B | 53.9 | 48.9 | 53.4 | 20.8 | 22.3 |
| Qwen3-14B | 57.7 | 51.8 | 56.2 | 21.8 | 24.6 |

Table 5: Average accuracy on ContextMATH. "w/o" denotes directly solving the scenario with the reasoning model.

## 6 CONCLUSION

In this paper, we investigate LLMs' ability to perform *contextual mathematical reasoning*, where the mathematical core must be extracted from narrative context before solving. Through the ContextMATH benchmark, we shows that models which perform strongly on abstract math exhibit large drops on contextual variants, with errors dominated by problem formulation. Our analyses identify formulation and reasoning as two complementary bottlenecks that jointly constrain performance. Although these processes may intertwine in complex problems, our controlled framework provides a first step toward disentangling them and demonstrates that formulation constitutes a key obstacle. We further show that end-to-end training with contextual data improves robustness, while directly training a dedicated formulation model is ineffective. Overall, our study establishes contextual mathematical reasoning as a central unsolved capability of LLMs and highlights it as a key frontier for progress toward reliable real-world applications.

## ACKNOWLEDGEMENT

The work described in this paper is substantially supported by a grant from the Direct Grant of Faculty of Engineering, The Chinese University of Hong Kong (Project Code: 4055275).

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

## A  APPENDIX

### ETHICS STATEMENT

This work studies LLMs through the lens of contextual mathematical reasoning. All data used are derived from publicly available math benchmarks (AIME and MATH-500), with additional scenario variants generated automatically and verified for mathematical equivalence. No private or sensitive data were used. We evaluate only open-source and commercially available models, and all experiments are conducted under non-interactive settings without user data collection. While our results reveal limitations of current LLMs, these findings are intended to encourage more rigorous evaluation and robust model design, not to promote unsafe deployment. We will release our benchmark openly to facilitate reproducibility and further research, subject to the same usage terms as the underlying datasets.

### REPRODUCIBILITY STATEMENT

We take reproducibility seriously in this work. The construction pipeline of **ContextMATH**, including all generation, verification, and revision prompts, is detailed in Section 3 and Appendix A.1. Evaluation setups are described in Appendix A.2, with complete accuracy results for open-source models reported in Appendix A.3, and results for proprietary models in Table 1. The analysis of formulation metrics is presented in Section 4, with full breakdowns in Appendix 7. Implementation details of all training experiments are provided in Appendix A.6. We will release our benchmark publicly, ensuring that independent researchers can fully reproduce our benchmark, analyses, and training results.

### THE USE OF LARGE LANGUAGE MODELS (LLMS)

In preparing this paper, we made limited use of LLMs to support the writing process. Specifically, LLMs were applied to check grammar, improve clarity, and polish the style of drafts. All technical content, experimental design, analysis, and conclusions are our own work, and the role of LLMs was restricted to language refinement.

### A.1  PROMPTS FOR DATA CONSTRUCTION

This section provides the **generation**, **verification**, and **revision** prompts used to transform original AIME and MATH problems into Scenario Grounding (SG; see sections A.1.1 to A.1.3) and Complexity Scaling (CS; see sections A.1.4 to A.1.6) versions.

### A.1.1  SCENARIO GROUNDING: GENERATION PROMPT

---

**SG Scenario Generation Prompt**

Your Goal: Convert an abstract math problem into a concrete, real-world story. The new story must be mathematically identical to the original. You can draft the story first, then use the following steps to ensure accuracy and format your final output correctly.

[Step 1] Map All Mathematical Components
List every single mathematical element from the original problem: numbers, variables, shapes, and operations. For each, assign a specific real-world analogue. Do not omit any numbers or symbols.

Bad Mapping (Vague): x → initial quantity
Good Mapping (Specific): x → The initial number of barrels of oil
Bad Mapping: sqrt(2) → a multiplier
Good Mapping: sqrt(2) → A special efficiency boost calculated as the square root of 2

---

---

**SG Scenario Generation Prompt (continued)**

Output Format for Step 1:
[Step 1] [Mathematical Element] → [Specific Real-World Analogue]
...

[Step 2] Define the Real-World Rules of Interaction
Combine the analogues from Step 1 into a coherent narrative context (e.g., engineering, physics, logistics). Explicitly define how the mathematical operations translate into actions or rules within your story. This step bridges the gap between individual components and the final narrative.

Example Math: f(x) = 2x + 3
Example Rule: "The output of a machine for any given input is found by doubling the input value and then adding a fixed calibration offset of 3."

Output Format for Step 2:
[Step 2] [Mathematical Operations] → [Specific Real-World Rule]
...

[Step 3] Write the Final Problem
Combine the elements from the previous steps into a concise story. Strict Requirements:
- Perfectly Equivalent: All numbers, relationships (like perpendicularity), and constraints from the original problem must be present and unambiguous in your story.
- No Math Language: Avoid mathematical terms (triangle, variable, x, cos) and symbols. Use descriptive names (a triangular field, the initial investment, a cosine-wave).
- Clear Question: The final question must ask for the same value as the original (with the same output format).

Output Format for Step 3:
[Step 3] Question: [The complete, final real-world problem in plain text.]

Now, apply this process to the following problem:
{original_problem_placeholder}

---

### A.1.2 SCENARIO GROUNDING: VERIFICATION PROMPT

---

**SG Scenario Verification Prompt**

You are a quality checker. Your task is to verify if a real-world story is a correct and clear representation of an original math problem.

Your Primary Goal:
Ensure perfect mathematical equivalence in a clear and concise narrative. The story does not need to be a hyper-realistic scientific model; it only needs to be a coherent and unambiguous scenario.

Please review the provided materials by considering these key questions:
1. Mathematical Integrity: Does the story perfectly preserve all numbers, relationships (e.g., perpendicularity, equality), and operations from the original problem? Is anything missing or changed?
2. Clarity & Language: Is the story easy to understand? Does it successfully avoid mathematical jargon and symbols, using clear descriptions instead? Is the final question unambiguous?
3. Conciseness: Is the story direct and to the point, without unnecessary details that overcomplicate the problem?

---

---

**SG Scenario Verification Prompt (continued)**

Your Output:
Please provide your review as a brief summary of your findings. After your summary, conclude with the mandatory [Overall Assessment] line.

Here is the output Format:
[Your findings] [Overall Assessment] - [Pass/Fail]

Here is the input for your review:
1. Original Math Problem {original_problem_placeholder}
2. Conversion Mappings:
2.1 Concept Mappings: {concept_mappings_list_placeholder}
2.2 Relationship and Attribute Mappings: {relationship_mappings_list_placeholder}
3. Real-World Problem: {real_world_problem_text_placeholder}

---

### A.1.3 SCENARIO GROUNDING: REVISION PROMPT

---

**SG Scenario Revision Prompt**

Carefully analyze the following feedback to revise your previous response to address all problems.

Here is the feedback:
{feedback_placeholder}

Output your revised version in plain text in the following format:
[Step 1]
[Mathematical Element] → [Specific Real-World Analogue]
...
[Step 2]
[Mathematical Operations] → [Specific Real-World Rule]
...
[Step 3]
Question: [The complete, final real-world problem in plain text.]

---

### A.1.4 COMPLEXITY SCALING: GENERATION PROMPT

---

**CS Scenario Generation Prompt**

Task: Enhancing Math Problem Difficulty through Contextual Embedding

Your goal is to increase the difficulty of a given mathematical problem's real-world version, while preserving its core mathematical structure and the integrity of its real-world mapping. Employ the following strategies:

—

1. Maintain Core Alignment
Ensure all mathematical quantities, variables, and relationships directly correspond to elements within the real-world context. The underlying mathematical problem must remain solvable and equivalent to the original.

—

CS Scenario Generation Prompt (continued)

2. Embed Conditions through Layered Obfuscation
Replace direct mathematical statements and values with more natural, layered, and context-rich descriptions that require deductive reasoning from the solver.

* A. Contextualized Numerical Properties (Disguising Specific Values):
    * This technique involves disguising specific numerical values by describing them through a property they possess within a realistic, tangible scenario. The goal is for the solver to derive the original number based solely on the provided context, using common knowledge or fundamental mathematical principles. Avoid creating custom or overly obscure scenarios that would require information beyond general understanding. Apply this method only when it naturally fits the number in question; don't force its application.
    * Example (for the value '4' hours): Rather than stating "lasts 4 hours," describe it as "lasts a number of hours equivalent to the unique combinations on the drone's two-digit pre-flight security keypad, where each digit can only be '1' or '2'." (This naturally leads to $2 \times 2 = 4$ combinations/hours).
    * Example (for the value '2'): "The unique positive whole number that, if you square it, then subtract one time itself, and finally subtract two, the result is zero."
    * Example (for the value '16'): "smallest integer such that C! is divisible by $2^{15}$."
* B. Relational & Functional Reverse Engineering (Disguising Variables/Formulas):
    * If the original problem involves analyzing a function with specific constants (e.g., $f(x) = (x - A)(x - B)...$), make these constants unknown (e.g., describe them as "labels are blurred,", "calibration is missing", etc.).
    * Provide a minimal set of realistic data points or observed behaviors sufficient for the solver to uniquely determine these unknown constants or the full function's form.
    * Example: Instead of giving $f(x) = \frac{(x-18)(x-72)...}{x}$, describe it as a system's performance, providing $P(22)$ and $P(33)$ values, requiring deduction of the hidden constants 18 and 72.
* C. Structural & Conceptual Reinterpretation (Disguising Geometric/Abstract Relations):
    * For geometric problems or abstract relationships, rephrase mathematical properties in terms of tangible real-world structures or processes.
    * Example (Symmetry): Describe a figure's symmetry as "a balanced design around a central axis" or "perfectly mirrored components." Then, we only describe half of the plot.
    * Example (Transformations): If there are scaling or rotation relationships, we can only describe a smaller prototype that is a scaled replica of other graphics.
    * Example (Equivalent Constraints): Reinterpret mathematical constraints using scenario-specific terms. For example, "a movable point lies along the perpendicular bisector of AC" instead of $AL = CL$. Another example is translating $x_k + \frac{1}{x_k}$ to $2\cos\theta_k$, if we let $x_k = e^{i\theta_k}$.

—

3. Introduce Plausible, Irrelevant Information
Incorporate details that are contextually relevant to the scenario but mathematically extraneous to the problem's solution. This increases cognitive load and realism. Example: For a drone problem, mention its battery capacity, wing span, payload weight, or the color of the packages being delivered, provided these details do not influence the distance-speed-time calculations.

—

4. Language Refinement and Simplicity:
Make sure all descriptions are concise and clear. Avoid redundancy and unnecessary repetition.

> **CS Scenario Generation Prompt (continued)**
>
> Avoid using mathematical variables (such as x, k, A, B, etc.) directly in the situation description, but give them specific real-world meanings (such as temperature, size, threshold, etc.). —
>
> Output Format:
> First think about how to apply these strategies to the math problem. Then output the enhanced problem in plain text after "Enhanced Contextual Math Problem:".
>
> —
>
> Here's the input:
> - Original Math Problem: {original_problem_placeholder}
> - Real-World Version: {real_world_problem_text_placeholder}

### A.1.5 COMPLEXITY SCALING: VERIFICATION PROMPT

> **CS Scenario Verification Prompt**
>
> You are tasked with verifying the quality and accuracy of a real-world version of an abstract math problem.
>
> Issue Types Defined:
>
>   * EQUIVALENCE_ERROR: The enhanced problem's mathematical core is not equivalent to the original, or it's unsolvable. Crucially, you must solve any nested or abstract sub-problems to confirm the derived mathematical information uniquely matches the original problem's data.
>   * Example Suggestion: "The definition for 'standard mission duration' should lead to a value of 4. Consider linking it to a 2 x 2 matrix operation if the context allows."
>
>   * REALISM_ERROR: The context or embedded condition feels contrived, unnatural, or like a "brain teaser" and isn't a plausible real-world scenario.
>   * Example Suggestion: "Instead of an overly abstract or obscure derivation for a value (e.g., a specific dimension or time), consider linking it to a more common measurement, a simple calculation based on widely known facts, or a basic geometric property that naturally fits the scenario."
>
>   * CONCISENESS_ERROR: The problem statement is unnecessarily long, verbose, or unclear.
>   * Example Suggestion: "Condense the description of the speed increment's derivation by directly stating the polynomial's property more concisely."
>
>   * FORMAT_ERROR: The output doesn't follow specified formatting (e.g., uses math variables, incorrect header, ask for different value).
>   * Example Suggestion: "Avoid using variable m in the context."
>
>   * OTHER: Any other issues.
>
> Review Output Format:
> Start your output with an `ASSESSMENT` status. If the `ASSESSMENT` is `FAIL`, you must provide `FEEDBACK` in the specified parseable format.
>
> [Overall Assessment]
> - [Pass/Fail]

> **CS Scenario Verification Prompt (continued)**
>
> If ASSESSMENT is FAIL, provide FEEDBACK like this:
>
> [FEEDBACK]
>
> - ISSUE_TYPE:
> [EQUIVALENCE_ERROR/REALISM_ERROR/...]
>     - DESCRIPTION: [Concise description of the specific issue.]
>     - LOCATION: [Optional: Specific phrase or section from the problem.]
> - ISSUE_TYPE: [Another ISSUE_TYPE if applicable]
>     - DESCRIPTION: [Another concise description.]
>
> Here is the input:
> - Original Math Problem {original_problem_placeholder}
> - Real-World Problem Scenario: {real_world_problem_text_placeholder}

### A.1.6 COMPLEXITY SCALING: REVISION PROMPT

> **CS Scenario Revision Prompt**
>
> Carefully analyze the following feedback to revise your previous response to address all problems.
>
> Here is the feedback:
> {feedback_placeholder}
>
> Output Format:
> Output the enhanced problem in plain text after "Enhanced Contextual Math Problem:".

### A.2 MODEL EVALUATION DETAILS

All open-source models tested were sourced from the Hugging Face Hub. We used the optimal generation parameters (e.g., temperature, top_p, top_k) for each model, following the configurations recommended in their respective papers or official GitHub repositories. The maximum response length was set to 32,768, or capped at the model's maximum supported length if shorter.

All proprietary models were evaluated using their official APIs or publicly available client interfaces. Unless specified otherwise, inference was conducted using greedy decoding (temperature = 0) for reproducibility (Shi et al., 2024), with a maximum response length of 32,768 tokens.

Some proprietary models were accessed via their web or client interfaces using default parameters during specific time windows. These include **Copilot**, **Grok3**, and **Gemini** (2.5 flash and 2.5 pro), which were evaluated for AIME 2024 between June 4th and June 11th, 2025, and for AIME 2025 between June 18th and June 24th, 2025.

For models accessed via API, we used their specified versions. These include **Doubao-1.5-thinking-pro** (version `2025-04-15`, 16k max tokens), **Qwen-max** (version `qwen-max-latest` on `2025-04-09`, 8k max tokens), **QwQ-plus** (version `qwq-plus-latest` on `2025-03-05`, 8k max tokens), **DeepSeek-R1** (version `250528`, 16k max tokens), **gpt-4o-mini** (`2024-07-18`), **o1-mini** (`2024-09-12`), **gpt-4o** (`2024-08-06`), **gpt-4.1-mini** (`2025-04-14`), **gpt-4.1-nano** (`2025-04-14`), and **o3** (`2025-04-16`).

`o1-mini` and `GPT-5` were accessed via shared endpoints that only support a temperature of 1.0.

### A.3 FULL RESULTS FOR OPEN-SOURCE MODELS

Table 6 reports the complete ContextMATH results for all evaluated open-source models, including: Qwen3-0.6B/1.7B/4B/8B/14B/32B (Yang et al., 2025a), Qwen2.5-Math-1.5/7B/14B, Qwen2.5-

| Model | AIME2024 (%) | | | AIME2025 (%) | | | Math-500 (%) | |
|---|---|---|---|---|---|---|---|---|
| | Ori | SG | CS | Ori | SG | CS | Ori | SG |
| **≤4B** | | | | | | | | |
| Qwen3-0.6B | 7.9 | 6.2 (-21%) | 0.4 (-95%) | 15.4 | 5.4 (-65%) | 2.5 (-84%) | 62.4 | 42.6 (-32%) |
| Qwen2.5-Math-1.5B | 11.7 | 2.5 (-79%) | 0.4 (-96%) | 4.6 | 0.8 (-82%) | 0.0 (-100%) | 32.5 | 25.8 (-21%) |
| ↪R1-Distil-Qwen-1.5B | 28.3 | 20.0 (-29%) | 7.1 (-75%) | 19.6 | 9.6 (-51%) | 5.8 (-70%) | 74.0 | 57.6 (-22%) |
| ↪DeepScaleR-1.5B-Preview | 41.7 | 23.3 (-44%) | 7.5 (-82%) | 30.4 | 15.4 (-49%) | 7.5 (-75%) | 80.1 | 64.2 (-20%) |
| ↪DeepMath-1.5B | 38.3 | 24.6 (-36%) | 10.8 (-72%) | 28.3 | 13.8 (-51%) | 5.0 (-82%) | 81.9 | 65.4 (-20%) |
| ↪Still-3-1.5B-Preview | 38.3 | 18.8 (-51%) | 7.9 (-79%) | 24.6 | 11.2 (-54%) | 2.5 (-90%) | 75.8 | 59.7 (-21%) |
| ↪OpenMath-Nemotron-1.5B | 62.5 | 34.2 (-45%) | 14.6 (-77%) | 50.4 | 21.2 (-58%) | 11.2 (-78%) | 87.3 | 70.1 (-20%) |
| ↪DeepMath-Omn-1.5B | 62.9 | 33.8 (-46%) | 13.8 (-78%) | 55.8 | 20.8 (-63%) | 12.5 (-78%) | 87.1 | 73.0 (-16%) |
| Qwen3-1.7B | 46.2 | 29.6 (-36%) | 12.5 (-73%) | 34.6 | 24.6 (-29%) | 11.2 (-67%) | 84.1 | 67.4 (-20%) |
| Qwen3-4B | 70.4 | 52.5 (-25%) | 34.6 (-51%) | 64.2 | 39.6 (-38%) | 33.8 (-47%) | 90.9 | 78.8 (-13%) |
| **7B/8B** | | | | | | | | |
| Qwen2.5-7B | 5.8 | 4.6 (-21%) | 1.7 (-71%) | 3.3 | 0.0 (-100%) | 0.4 (-87%) | 43.9 | 30.4 (-31%) |
| ↪Open-Reasoner-Zero-7B | 15.8 | 12.5 (-21%) | 6.2 (-61%) | 14.6 | 7.1 (-51%) | 2.5 (-83%) | 69.3 | 53.0 (-23%) |
| ↪DeepMath-Zero-7B | 18.8 | 7.5 (-60%) | 5.8 (-69%) | 15.8 | 9.2 (-42%) | 2.1 (-87%) | 74.2 | 59.6 (-20%) |
| Qwen2.5-Math-7B | 10.8 | 6.7 (-38%) | 1.7 (-85%) | 5.0 | 3.3 (-33%) | 0.8 (-83%) | 44.8 | 36.7 (-18%) |
| ↪OpenMath-Nemotron-7B | 72.9 | 52.1 (-29%) | 30.0 (-59%) | 60.0 | 40.4 (-33%) | 29.2 (-51%) | 90.5 | 78.5 (-13%) |
| ↪R1-Distil-Qwen-7B | 48.8 | 40.0 (-18%) | 23.3 (-52%) | 41.7 | 22.5 (-46%) | 15.0 (-64%) | 87.4 | 73.9 (-15%) |
| ↪AceMath-RL-Nemotron-7B | 69.2 | 48.8 (-30%) | 32.9 (-52%) | 54.2 | 26.7 (-51%) | 22.5 (-58%) | 89.9 | 79.0 (-12%) |
| ↪Qwen2.5-Math-7B-SRL-Zero | 16.2 | 6.7 (-59%) | 0.8 (-95%) | 4.6 | 3.3 (-27%) | 0.0 (-100%) | 49.6 | 33.4 (-33%) |
| ↪Qwen2.5-Math-7B-SRL | 20.8 | 14.2 (-32%) | 9.2 (-56%) | 17.5 | 10.4 (-40%) | 2.9 (-83%) | 70.7 | 58.3 (-18%) |
| ↪DeepMath-Zero-Math-7B | 30.4 | 18.8 (-38%) | 9.2 (-70%) | 22.1 | 15.8 (-28%) | 7.1 (-68%) | 77.4 | 63.8 (-18%) |
| ↪Qwen2.5-Math-7B-Oat-Zero | 32.5 | 18.8 (-42%) | 2.9 (-91%) | 11.2 | 9.2 (-18%) | 0.4 (-96%) | 66.2 | 44.2 (-33%) |
| ↪Eurus-2-7B-PRIME-Zero | 20.0 | 13.3 (-33%) | 6.7 (-67%) | 6.7 | 10.0 (-50%) | 0.0 (-100%) | 59.9 | 39.7 (-34%) |
| ↪Eurus-2-7B-SFT | 6.7 | 3.3 (-50%) | 3.3 (-50%) | 3.3 | 0.0 (-100%) | 0.0 (-100%) | 50.0 | 38.5 (-23%) |
| ↪Eurus-2-7B-PRIME | 20.0 | 13.3 (-33%) | 6.7 (-67%) | 10.0 | 3.3 (-67%) | 3.3 (-67%) | 68.3 | 57.6 (-16%) |
| R1-Distil-Llama-8B | 43.3 | 29.2 (-33%) | 10.8 (-75%) | 30.4 | 17.5 (-42%) | 11.2 (-63%) | 81.6 | 65.6 (-20%) |
| Qwen3-8B | 73.8 | 61.7 (-16%) | 42.9 (-42%) | 64.6 | 48.3 (-25%) | 35.8 (-45%) | 91.0 | 81.0 (-11%) |
| ↪DeepSeek-R1-0528-Qwen3-8B | 75.0 | 55.0 (-27%) | 39.6 (-47%) | 65.8 | 48.3 (-27%) | 32.9 (-50%) | 90.7 | 77.2 (-15%) |
| ↪AReaL-boba-2-8B | 74.2 | 58.3 (-21%) | 41.7 (-44%) | 67.9 | 47.9 (-29%) | 37.1 (-45%) | 91.8 | 82.0 (-11%) |
| Qwen2.5-Math-7B-Instruct | 11.2 | 6.2 (-44%) | 2.5 (-78%) | 9.2 | 5.0 (-45%) | 0.0 (-100%) | 72.0 | 54.0 (-25%) |
| **14B** | | | | | | | | |
| Qwen2.5-14B | 6.2 | 3.8 (-40%) | 1.7 (-73%) | 3.3 | 1.7 (-50%) | 0.0 (-100%) | 48.5 | 34.9 (-28%) |
| ↪R1-Distil-Qwen-14B | 67.5 | 47.1 (-30%) | 35.4 (-48%) | 50.8 | 26.2 (-48%) | 25.8 (-49%) | 89.5 | 76.9 (-14%) |
| ↪OpenMath-Nemotron-14B | 73.8 | 51.7 (-30%) | 42.1 (-43%) | 63.8 | 42.5 (-33%) | 29.2 (-54%) | 90.8 | 80.8 (-11%) |
| Qwen3-14B | 80.0 | 64.6 (-19%) | 50.8 (-36%) | 72.9 | 49.2 (-33%) | 42.1 (-42%) | 92.6 | 81.9 (-12%) |
| ↪AReaL-boba-2-14B | 82.9 | 65.8 (-21%) | 53.8 (-35%) | 73.3 | 52.1 (-29%) | 39.2 (-47%) | 91.9 | 82.9 (-10%) |
| Phi-4-reasoning-plus | 80.4 | 60.4 (-25%) | 52.9 (-34%) | 71.7 | 55.4 (-22%) | 39.6 (-45%) | 92.6 | 83.1 (-10%) |
| **≥32B** | | | | | | | | |
| Qwen2.5-32B | 11.2 | 6.7 (-41%) | 3.3 (-70%) | 3.8 | 2.9 (-22%) | 0.0 (-100%) | 45.2 | 37.8 (-16%) |
| ↪OpenMath-Nemotron-32B | 57.1 | 42.5 (-26%) | 27.9 (-51%) | 52.1 | 34.6 (-34%) | 27.5 (-47%) | 75.8 | 61.8 (-18%) |
| ↪R1-Distil-Qwen-32B | 69.6 | 52.5 (-25%) | 39.2 (-44%) | 56.2 | 39.6 (-30%) | 30.0 (-47%) | 89.4 | 78.9 (-12%) |
| ↪Open-Reasoner-Zero-32B | 43.8 | 30.4 (-30%) | 25.4 (-42%) | 32.9 | 25.4 (-23%) | 15.0 (-54%) | 84.3 | 75.1 (-11%) |
| Qwen3-32B | 81.2 | **67.9** (-16%) | 57.1 (-30%) | 70.0 | 54.4 (-22%) | **45.0** (-36%) | 92.1 | 82.7 (-10%) |
| ↪AReaL-boba-2-32B | 81.7 | 65.4 (-20%) | 58.3 (-29%) | 77.1 | 55.0 (-29%) | 43.8 (-43%) | 92.3 | 82.9 (-10%) |
| QwQ-32B | 80.4 | 58.3 (-27%) | 53.3 (-34%) | 66.2 | 53.3 (-20%) | 39.2 (-41%) | 92.5 | 82.9 (-10%) |
| Qwen2.5-32B-Instruct | 15.4 | 14.2 (-8%) | 5.4 (-65%) | 15.8 | 9.2 (-42%) | 1.7 (-89%) | 70.3 | 54.5 (-23%) |
| ↪s1.1-32B | 58.8 | 47.9 (-18%) | 38.3 (-35%) | 46.7 | 36.2 (-22%) | 32.1 (-31%) | 88.8 | 77.9 (-12%) |
| DeepSeek-R1-Distill-Llama-70B | 65.4 | 48.8 (-25%) | 38.8 (-41%) | 50.0 | 38.8 (-22%) | 29.2 (-42%) | 89.8 | 77.3 (-14%) |
| Qwen2.5-Math-72B-Instruct | 19.2 | 13.8 (-28%) | 6.7 (-65%) | 14.6 | 7.5 (-49%) | 3.3 (-77%) | 74.7 | 60.5 (-19%) |

Table 6: Full results of open-source models on ContextMATH. Models are grouped by parameter scale. Parentheses indicate the relative change from Ori, with larger drops highlighted by deeper red. Arrows (↪) denote variants obtained from the model listed directly above.

Math-7B/72B-Instruct (Yang et al., 2024), DeepSeek-R1-Distil-Qwen-1.5B/7B/14B/32B, R1-Distil-Llama-8B/70B,DeepSeek-R1-0528-Qwen3-8B (Guo et al., 2025), DeepScaleR-1.5B-Preview (Luo et al., 2025), Still-3-1.5B-Preview (Team, 2025b), Phi-4-reasoning-plus (Abdin et al., 2025b), Qwen2.5-7B/14B/32B, Qwen2.5-32B-Instruct (Team, 2024), Open-Reasoner-Zero-7B/32B (Hu et al., 2025), DeepMath-1.5B, DeepMath-Zero-7B, DeepMath-Omn-1.5B (He et al., 2025), OpenMath-Nemotron-1.5B/7B/14B/32B (Moshkov et al., 2025), AceMath-RL-Nemotron-7B (Liu et al., 2024), Qwen2.5-Math-7B-SRL-Zero, Qwen2.5-Math-7B-SRL (Zeng et al., 2025), Qwen2.5-Math-7B-Oat-ZeroL (Liu et al., 2025), Eurus-2-7B-PRIME-Zero, Eurus-2-7B-SFT, Eurus-2-7B-PRIME (Cui et al., 2025), AReaL-boba-2-8B/14B/32B (Fu et al., 2025), Phi-4-reasoning-plus (Abdin et al., 2025a), QwQ-32B (Team, 2025a), and s1.1-32B (Muennighoff et al., 2025). Across architectures and training variants, accuracy drops sharply on both SG and CS tasks compared to the original problems, with the CS setting consistently causing the largest degradation. Larger models generally achieve higher accuracy, yet significant performance gaps remain. We also observe that while an initial stage of supervised fine-tuning tends to improve robustness, additional specialized tuning (e.g., RL or further SFT) often yields little benefit for contextual tasks (Luo et al., 2023; Shi et al., 2023; Shao et al., 2024; Wang et al., 2024; Meng et al., 2024; Yang et al., 2025b; Guo et al., 2025) Overall, these expanded results provide a comprehensive view of model behavior that underlies the key findings discussed in Section 3.4.

## A.4 Reasoning Error Example of DeepSeek R1

We provide an example in A.4.1 to illustrate a typical failure mode of `DeepSeek R1`. The model's algebraic reasoning was largely correct, but it misinterpreted how geometric constraints in the scenario map back to the original problem, leading to an incorrect answer. This case highlights how formulation errors can propagate even when subsequent reasoning is valid.

### A.4.1 Case Study: Misinterpreting Geometric Constraints

> **An example of R1 reasoning error**
>
> **Original question:**
> Rectangles $ABCD$ and $EFGH$ are drawn such that $D, E, C, F$ are collinear. Also, $A, D, H, G$ all lie on a circle. If $BC = 16, AB = 107, FG = 17$, and $EF = 184$, what is the length of $CE$?
>
> **Generated Scenario:**
> At a large construction site, two rectangular platforms are being assembled for a robotics demonstration. The first platform, called the Alpha Pad, is 107 meters long and 16 meters wide. The second platform, the Echo Deck, is 184 meters long and 17 meters wide.
> The platforms are being arranged in a very specific layout:
> The back-right corner of the Alpha Pad, the front-left corner of the Echo Deck, the front-right corner of the Alpha Pad, and the front-right corner of the Echo Deck all lie along this straight track.
> Meanwhile, the back-left and the back-right corners of both platforms are all positioned along the edge of a circular fountain that marks the center of the plaza.
> Given this setup, how far apart are the front-right corner of the Alpha Pad and the front-left corner of the Echo Deck?
>
> **R1 response:**
> The platforms are arranged such that the back-right corner of the Alpha Pad (BR_A), the front-left corner of the Echo Deck (FL_E), the front-right corner of the Alpha Pad (FR_A), and the front-right corner of the Echo Deck (FR_E) lie on a straight track. Additionally, the back-left and back-right corners of both platforms lie on a circular fountain centered at the plaza's center.

> **An example of R1 reasoning error (continued)**
>
> The Alpha Pad is 107 meters long and 16 meters wide, and the Echo Deck is 184 meters long and 17 meters wide. The straight track is treated as the x-axis in a coordinate system, with the fountain center at (h, k) and radius r. The back corners on the fountain and the points on the track are used to determine the positions.
> For the Alpha Pad:
> - BR_A is at (0, 0).
> - FR_A is at (107, 0), as the distance along the length is 107 meters.
> - BL_A is at (0, 16), assuming the platform is above the track ($y > 0$).
> This gives k = 8 from the circle conditions, and $r^2 = h^2 + 64$.
> For the Echo Deck:
> - FL_E is at (b, 0).
> - FR_E is at (b + 17, 0), as the width is 17 meters. | **Incorrect mathematical interpretation**
> - BL_E is at (b, 184), assuming the platform is above the track ($y > 0$).
>
> **The subsequent reasoning process is correct, but based on wrong conditions.**
>
> **GPT-5 analysis:**
> Primary error type: Incorrect mathematical interpretation (with a secondary logical/rounding issue at the end).
> What went wrong:
> 1) Misassigning which side of the Echo Deck lies along the straight track. ...
> 2) Misassigning which Echo Deck side forms the chord on the circle. ...
> 3) Additional logical issue (unjustified "nice fraction".) ...
> How this led to the wrong answer:
> - By putting 17 along the track instead of 184 and using a 17-length chord for the circle instead of 184, the computed b (the x-position of FL_E) shifted incorrectly, producing $|107 - b| \approx 60.5$, rather than the correct $CE = 104$. The mistake is structural (wrong mapping of sides to the line and circle), not just arithmetic.

| Model | Formulation Accuracy (%) | | | | | Formulation Necessity (%) | | | | | Formulation Sufficiency (%) | | | | |
|---|---|---|---|---|---|---|---|---|---|---|---|---|---|---|---|
| | AIME24 | | AIME25 | | MATH | AIME24 | | AIME25 | | MATH | AIME24 | | AIME25 | | MATH |
| | SG | CS | SG | CS | SG | SG | CS | SG | CS | SG | SG | CS | SG | CS | SG |
| R1-Distil-Qwen-1.5B | 57.5 | 40.0 | 49.2 | 31.7 | 62.3 | 61.4 | 43.8 | 41.7 | 75.0 | 70.6 | 17.5 | 10.1 | 9.1 | 15.9 | 65.0 |
| R1-Distil-Qwen-7B | 60.0 | 55.8 | 53.3 | 40.8 | 75.7 | 67.8 | 66.2 | 63.2 | 58.3 | 82.9 | 46.3 | 31.1 | 26.7 | 22.2 | 81.1 |
| R1-Distil-Qwen-14B | 74.2 | 63.3 | 53.3 | 50.0 | 85.7 | 80.8 | 71.7 | 78.3 | 81.3 | 90.1 | 53.7 | 42.2 | 31.7 | 42.3 | 80.8 |
| R1-Distil-Qwen-32B | 77.5 | 65.8 | 59.2 | 60.0 | 87.3 | 80.1 | 77.0 | 69.8 | 78.0 | 91.8 | 51.7 | 45.9 | 46.1 | 39.4 | 83.5 |
| Qwen3-0.6B | 46.7 | 29.2 | 45.8 | 35.0 | 57.4 | 52.8 | 100.0 | 33.3 | 25.0 | 69.2 | 8.4 | 3.6 | 1.5 | 1.9 | 52.2 |
| Qwen3-1.7B | 66.7 | 47.5 | 60.0 | 53.3 | 82.5 | 73.7 | 84.5 | 68.4 | 83.3 | 90.1 | 32.4 | 24.8 | 27.6 | 14.1 | 73.8 |
| Qwen3-4B | 74.2 | 55.0 | 51.7 | 42.5 | 84.8 | 91.1 | 79.9 | 63.7 | 71.0 | 90.3 | 65.5 | 48.8 | 50.7 | 57.8 | 83.6 |
| Qwen3-8B | 81.7 | 74.2 | 66.7 | 60.0 | 86.3 | 89.8 | 83.6 | 77.3 | 77.2 | 91.3 | 66.0 | 49.3 | 54.2 | 47.7 | 86.3 |
| Qwen3-14B | 77.5 | 68.3 | 66.7 | 62.5 | 87.1 | 82.1 | 74.8 | 81.0 | 95.8 | 90.4 | 64.6 | 57.4 | 62.4 | 62.0 | 85.2 |
| Qwen3-32B | 84.2 | 70.0 | 72.5 | 59.2 | 89.3 | 80.5 | 74.0 | 86.2 | 76.6 | 92.0 | 66.4 | 59.6 | 57.5 | 55.7 | 85.0 |
| gpt-5 | 93.3 | 76.7 | 80.0 | 66.7 | 90.5 | 96.0 | 79.2 | 83.3 | 75.0 | 94.5 | 85.7 | 82.6 | 83.3 | 75.0 | 86.9 |

Table 7: Problem formulation performance comparison across different models. Formulation Accuracy denotes the proportion of cases in which a model correctly translates a scenario into its mathematical formulation. Formulation Necessity quantifies the extent to which correct formulations are required for correct reasoning. Formulation Sufficiency evaluates whether correct formulations reliably lead to correct reasoning. Darker cell colors correspond to higher values.

## A.5 FULL RESULTS FOR PROBLEM FORMULATION ANALYSIS

Table 7 reports the complete formulation performance results. The finer granularity confirms the same overall trends: formulation accuracy decreases with problem difficulty, necessity consistently exceeds accuracy, and sufficiency improves with scale but remains lower overall. These detailed

results provide a comprehensive view of model behavior underlying the averaged metrics discussed in Section 4.2.

## A.6 IMPLEMENTATION DETAILS.

We fine-tuned the model in the full-parameter SFT setting on 4×A100 80GB GPUs. We adopted a per-device batch size of 1, accumulated over 32 steps, yielding an effective global batch size of 128. The model was optimized for 1200 steps with a cosine scheduler, 10% warmup, and a peak learning rate of 5e-5. Mixed-precision training with bf16 was enabled.

