# OpenReview forum: "From Abstract to Contextual: What LLMs Still Cannot Do in Mathematics"
_ICLR.cc/2026/Conference — ICLR 2026 Poster_

### Official Review · Reviewer_xYoD · 2025-10-25

**Soundness:** 2
**Presentation:** 2
**Contribution:** 2
**Rating:** 6
**Confidence:** 5

**Summary:**

This paper tackles a very relevant gap, i.e. models that perform well on abstract math often struggle when the same problems are presented in a realistic narrative. The authors build a benchmark by transforming existing items from AIME and a filtered MATH-500 subset into two contextualized variants. The first, Scenario Grounding (SG), wraps the same core math in a short story while keeping the solution unchanged. The second, Complexity Scaling (CS), conceals some explicit conditions behind small sub-problems, requiring the model to first identify what is needed before solving. They evaluate a large set of models and observe a clear drop from the original to SG and a further drop to CS. They also analyze where errors originate and introduce three formulation metrics (formulation accuracy, necessity, and sufficiency) to distinguish between “setting up the right math” and “doing the math.”. The authors test training strategies and find that simple SFT on scenario-style data gives measurable improvements without harming abstract performance.

I see the contribution as timely and practically important. The benchmark design, although not perfect, addresses a real need: testing whether models can extract the correct structure from a story. The formulation-vs-reasoning framing is helpful for analysis, and the training results give a simple recipe that others can try. The novelty is, however, moderate (a new benchmark plus analysis), but the empirical scope and the focus on formulation make it valuable to the community, provided the evaluation is strengthened.

**Strengths:**

The work isolates a phenomenon that has been widely reported, ie, strong models on abstract math often fail when conditions are embedded in text. The two-variant construction is simple and scalable, and the error analysis is quite illuminating, showing that setting up the math is frequently the key difficulty. I also appreciate that the authors do not stop at measurement, but also include initial training interventions that enhance contextual performance without compromising abstract skills. The paper is mostly clear, and the benchmark seems practical to adopt.

**Weaknesses:**

My main concerns are about measurement validity and fairness. The SG/CS transformations would benefit from multiple independent annotators and a quantitative audit to ensure equivalence and controlled change. The formulation metrics rely on an LLM judge. The human validation sample is eventually too small, and there is no check with a different judge family or a symbolic/exact solver, where feasible. The open-source vs proprietary comparison lacks confidence intervals to assess robustness. Contamination or near-duplicate checks are not presented, which matters for items derived from widely circulated math sets. The SFT pipeline filters scenarios using one solver, which can introduce selection bias and may not generalize.

**Questions:**

1) Could you report inter-annotator agreement and a small difficulty-parity audit for SG, and a clearer rubric for CS to show it does not change the task class?
2) Would you consider re-scoring a stratified subset with a different judge family, some exact/symbolic checks where possible, and a larger blinded human pool?
3) It would also help to provide confidence intervals or bootstraps, and to balance sampling across model groups (or clearly separate the summaries).
4) Please comment on step-level near-duplicate analysis and on the sensitivity of the SFT gains to the choice of solver used for filtering.

---

> ### Author Response · Authors · 2025-11-17
>
> Thank you for your positive assessment of our work's timeliness, practical importance, and analytical value. We appreciate that your questions directly correspond to the weaknesses raised, and we address each concern below.
>
> &nbsp;
>
> **[Q1]** Could you report inter-annotator agreement and a small difficulty-parity audit for SG, and a clearer rubric for CS to show it does not change the task class?
>
> **[A1]** We agree that the review protocol is critical for ensuring validity. Accordingly, we have implemented a rigorous multi-stage review process.
>
> 1.  Inter-annotator agreement and quantitative audit
>
> For inter-annotator agreement, our procedure involves three experts, all of whom have advanced degrees in Computer Science and backgrounds in competitive mathematics. They conduct independent reviews based on the following criteria:
>
> - Assessing the scenario for narrative plausibility and clarity, ensuring it introduces no unnecessary complexity or ambiguity.
> - Independently formulating an abstract math problem from the scenario to verify its solvability and mathematical equivalence to the original.
> - Testing the scenario on Gemini and GPT-5; if these models fail, the reviewer diagnoses whether the failure originates from an ambiguity in the problem description.
>
> A scenario is accepted only after passing all checks. Any identified issue flags the scenario for revision. In cases of disagreement, the reviewer with the strongest mathematical background leads a discussion to reach a consensus, which may trigger a regeneration or revision cycle.
>
> **All annotators reached a consensus on the validity of each final scenario.**
>
> 2. Difficulty-parity audit for SG
>
> We ensure difficulty parity for SG variants through two complementary approaches. First, in our **prompting strategy**, as shown in Appendix A.2.1, we explicitly require the model to:
>
> - "[Step 1] Map All Mathematical Components" (line 677), where mathematical elements are directly mapped to real-world objects one-by-one.
> - "[Step 2] Define the Real-World Rules of Interaction" (line 690), where mathematical relationships are mapped to reasonable real-world descriptions.
> - "[Step 3] Write the Final Problem" (line 704), which simply integrates all information.
>
> We specifically require that "The final question must ask for the same value as the original (with the same output format)" (line 710). Additionally, in the verification phase, we prompt the model to check "Is the story direct and to the point, without unnecessary details that overcomplicate the problem?" (line 741). These requirements prevent the introduction of additional difficulty.
>
> Second and more fundamentally, during **human review**, the primary criterion is to ensure mathematical equivalence and no information beyond the original problem is introduced - only mathematical concepts and relationships are replaced with contextualized natural language descriptions.
>
> 3. Clearer rubric for CS transformations
>
> For CS transformations, we also employ a two-layer approach through **prompting strategy and human review**. In prompting, as shown in Appendix A.2.4, we explicitly require:
> "The goal is for the solver to derive the original number based solely on the provided context, using common knowledge or fundamental mathematical principles. Avoid creating custom or overly obscure scenarios that would require information beyond general understanding. Apply this method only when it naturally fits the number in question; don't force its application." (lines 799-802)
>
> During human review, we apply the same standard, where experts must independently formulate the abstract math problem without tool assistance. Typical transformation examples include encoding 16 as "two raised to the power of four" and 12 as "the number of edges on a standard cuboid." Somewhat more complex cases involve replacing constants with equation systems, such as encoding $m=n^4+1$ as "Calibrations show: 1 rod spin yields 2 sculpture rotations; 2 rod spins yield 17 sculpture rotations. (i.e., $2=1^x+y, 17=2^x+y, x,y>0 → x=4, y=1$)". However, these sub-problems are self-contained and rely only on basic mathematical knowledge, requiring few additional step of independent reasoning to recover the original information.
>
> Additionally, we count the number of such transformations in the AIME 2024 and AIME 2025 CS sets, with averages of 2.4 and 2.3 transformations per problem respectively. We also emphasize that CS problems average 176 words (line 213). Therefore, we conclude that CS does not increase task difficulty nor introduce reading comprehension challenges. We will include these details in the revised version.

---

> > ### Comment · Reviewer_xYoD · 2025-11-25
> >
> > Thank you for the clarification. In light of the thorough responses, I will increase my score.

---

> > > ### Author Response · Authors · 2025-11-26
> > >
> > > Thank you for your positive feedback! We are glad that our rebuttal addresses your concerns. We will also address these concerns and incorporate all your suggestions in our paper. Thanks again for your insightful comments that help improve our work greatly.

---

> ### Author Response · Authors · 2025-11-17
>
> **[Q2]** Would you consider re-scoring a stratified subset with a different judge family, some exact/symbolic checks where possible, and a larger blinded human pool?
>
> **[A2]** Thank you for raising this important point. We are pleased to elaborate on the analysis of consistency across different judges and the human validation details.
>
> We conduct cross-judge validation using GPT-4.1 on all sampled solutions from Qwen3-14B and Qwen3-32B across all subsets (382 problems × 16 answers per problem per model × 2 models= 12,224 answers). The agreement rate between GPT-4.1 and o1-mini in assigning identical binary labels reaches **98.13%**, demonstrating the high reliability of our formulation metrics across different judge families.
>
> For the mathematical equivalence judgment by o1-mini (Section 4.2), we performed **a human check on 1,528 model outputs**. These were obtained by randomly sampling 2 outputs from each of the 16 generations per problem for Qwen3-14B and Qwen3-32B, across the AIME 2024, AIME 2025, and MATH-500 SG/CS sets (382 × 2 × 2=1,528). Both human and o1-mini judges assigned a binary label (0/1) for mathematical equivalence. The agreement rate, measured as the proportion of identical labels, was 94%, which we deem sufficient for our analysis.
>
> Regarding exact/symbolic checks, this approach is unfortunately not applicable in our setting. The formulated mathematical problems are open-ended, unstructured textual expressions generated by the models, not standardized symbolic inputs.
>
> We will provide full details of these validation analyses in the revised version.
>
> &nbsp;
>
> **[Q3]** It would also help to provide confidence intervals or bootstraps, and to balance sampling across model groups (or clearly separate the summaries).
>
> **[A3]** Thank you for this suggestion. To ensure we address it precisely, could you kindly specify which particular open-source vs. proprietary comparison you are referring to, and which experiments you feel would benefit from more balanced sampling across model groups?
>
> We would like to clarify that we have striven to maintain fairness in our evaluation design. For instance, in the formulation task (Table 3), all models, including GPT-5, were evaluated with 16 samples per problem. The only exception occurs in the reasoning task (Table 1), where some proprietary models were accessed via web interfaces while others were called through API, as noted in Appendix A.3. We will make this distinction clearer in the revised version by explicitly separating these proprietary models.
>
> Moreover, recognizing the difference in sampling strategies between open-source models (16 samples per problem) and proprietary models (single pass) for the reasoning task, we intentionally refrained from direct performance comparisons between the two groups in Section 3.4 (result analysis), instead discussing their trends separately.
>
> We are happy to provide any additional information and welcome further guidance on strengthening this aspect of our analysis.

---

> ### Author Response · Authors · 2025-11-17
>
> **[Q4]** Please comment on step-level near-duplicate analysis and on the sensitivity of the SFT gains to the choice of solver used for filtering.
>
> **[A4]** Regarding contamination and near-duplicate checks, could you please specify which particular part of our experiment concerns you? We would appreciate your suggestions on how to best implement near-duplicate checks in our context.
>
> Regarding solver selection for SFT data filtering, before constructing the SFT dataset, we analyzed approximately 500 original math problems (from DeepMath-103K) to evaluate different models as solvers (which also serve as annotators, with their solutions used as SFT labels). In terms of accuracy, Qwen3-32B achieves the best performance among open-source models (≈90%), even surpassing proprietary models such as o1-mini (≈70%). This observation is consistent with the findings presented in Tables 1 and 2 of our paper. While proprietary models such as DeepSeek-R1 and GPT-5 perform better, their cost for constructing large-scale SFT data is prohibitively high. This favorable balance between performance and cost efficiency led us to select Qwen3-32B.
>
> We also experimented with using o1-mini as the solver. However, the data filtered by o1-mini largely overlaps with the set filtered by Qwen3-32B (exhibiting >85% overlap). We were also concerned that using other models such as o1-mini might introduce lower-quality solutions, given their comparatively lower performance on the original problem set. Therefore, we ultimately used only Qwen3-32B for filtering.
>
> Regarding sensitivity, we acknowledge that models may have their own reasoning patterns, and selecting a specific model to build SFT data may introduce bias. However, we believe that for mathematical reasoning tasks, obtaining high-quality solutions is more important. In fact, our SFT strategy not only achieves significant benefits on our dataset but also improves performance on AMC23 (abstract mathematical reasoning) and Math-P (which tests generalization under data shifts), as shown in Table 4. This demonstrates that our SFT approach enhances fundamental mathematical capabilities. Furthermore, as Qwen3-32B is open-source, we believe this method offers a practical and accessible approach for application in other domains.

---

> ### Author Response · Authors · 2025-11-24
> **Follow up on the reviews**
>
> Thank you again for your constructive feedback and for taking the time to review our work.
>
> We hope our responses have addressed your concerns, and we would be grateful if you could consider increasing the score based on our clarifications.
>
> We remain available for any further questions.

---

### Official Review · Reviewer_qQ6S · 2025-10-31

**Soundness:** 3
**Presentation:** 3
**Contribution:** 2
**Rating:** 4
**Confidence:** 3

**Summary:**

This paper introduces CORE-MATH, a benchmark for evaluating LLMs on contextual math reasoning. The benchmark converts problems from AIME and MATH-500 into scenario-based narratives and more complex narratives with embedded sub-problems, Complexity Scaling. The authors evaluate 61 and proprietary open-source LLMs and find that performance drops sharply when moving from abstract to contextual settings. Through error analysis and targeted experiments, the main limitation is identified as incorrect problem formulation rather than failures in step-by-step reasoning. The paper also examines how fine-tuning and formulation-specific training can reduce these errors and shows gains, but a large gap remains. The work demonstrates that contextual math reasoning is still an open challenge for current LLMs.

**Strengths:**

- The research question is well motivated and the main finding is relevant. The paper provides a useful observation: the authors identify contextual complexity as a general bottleneck that limits current LLMs' reliability on multi-step reasoning.

- The paper provide insightful error analysis. The detailed breakdown of error types (Figure 2) demonstrates that formulation errors predominate across architectures in contextual settings, which is often overlooked in prior work.

**Weaknesses:**

- The paper builds several automatic annotation and categorization steps and relies on an LLM judge with only light human checks. The authors state that they use o1-mini to decide whether a model output is mathematically equivalent to the reference solution, and that manual checks on Qwen3-14B and Qwen3-32B show more than 90% agreement with human judgments. However, the paper does not report the exact sample sizes, selection protocol, prompts, or agreement statistics, e.g., Cohen's kappa, Pearson/Spearman correlation of scores, or confidence intervals. The heavy use of an LLM-as-judge without full validation details becomes a concern.

- The error analysis uses GPT-5 to assign error categories to outputs from other models, but the paper does not provide the prompts or templates that define each category. Also, there is no study of judge bias or stability, e.g., prompt ablations, temperature sweeps, and no human expert evaluation. A small human audit with inter-rater agreement would be useful.

- CORE-MATH draws from AIME-2024, AIME-2025, and a filtered subset of MATH-500, with each original item converted into two variants, SG and CS. This means the total pool starts from a relatively small number of sources. The paper probes robustness by adding two extra SG versions for AIME-2024 and reporting an averaged score, SG Avg@3, but it does not report standard deviation or standard error, and it does not repeat this check for CS or the other subsets. It is not clear whether that the results could be sensitive to specific paraphrases; Another concern is the limited scale in CORE-MATH, which makes results less convincing.

**Questions:**

- The proposed SFT setup helps but does not remove the drop under CS. The paper positions this as a first step, but the current training recipe and analysis do not yet show a clear path to solve the remaining gap. What suggestions do the authors have to further address this research gap, or how would the authors justify the limited improvement through their method?

- Typo: the caption reference "Figure 4" should be "Table 5."

---

> ### Author Response · Authors · 2025-11-17
>
> Thank you for your thorough review and for acknowledging our work’s relevance and the error analysis in our work. We appreciate your constructive feedback on areas needing further elaboration. We address each of your points below.
>
> &nbsp;
>
> **[W1] Lack of detailed validation for the LLM-as-Judge setup.**
>
> **[A1]** Thank you for raising this important point.
> For the mathematical equivalence judgment by o1-mini (Section 4.2), we performed a human check on 1,528 model outputs. These were obtained by randomly sampling 2 outputs from each of the 16 generations per problem for Qwen3-14B and Qwen3-32B, across the AIME 2024, AIME 2025, and MATH-500 SG/CS sets. Both human and o1-mini judges assigned a binary label (0/1) for mathematical equivalence. The agreement rate, measured as the proportion of identical labels, was 94%, which we deem sufficient for our analysis. The prompt used is provided below.
>
> Other LLM-judge uses were also validated:
>
> - Benchmark construction (Section 3.2): All scenarios generated and self-verified by o1-mini were subsequently verified by human experts.
> - Error categorization (Section 4.1): We will elaborate on the details in our response to Weakness 2 (A2) regarding the validation of GPT-5 for error categorization.
>
> While LLM judges may not achieve perfect human alignment, our validation confirms they provide reliable judgments for these specific tasks. This approach enabled scalable analysis of extensive reasoning traces, with human checks ensuring result validity.
> We will provide full details in the revised version.
>
>
>
> `Prompt for mathematical equivalence judgment`
> ```
> Role: You are an expert mathematician and logician. Your task is to determine whether two given math problems are mathematically equivalent.
>
> Definition:
> Two problems are mathematically equivalent if and only if:
>
> 1. They involve the same core mathematical concepts and operations.
> 2. The logical relationships and constraints between variables/entities are identical.
> 3. All numerical values and constants are the same, or any differences are purely superficial (e.g., renaming a variable from `x` to `y` throughout the entire problem).
> 4. They ask for the same final quantity or result, and solving one necessarily leads to the same answer as solving the other.
>
> Evaluation Criteria:
> Please analyze the equivalence based on the following aspects. The problems MUST be equivalent in ALL aspects to be deemed equivalent.
>
> - Mathematical Concepts: Are the underlying mathematical disciplines and fundamental concepts the same (e.g., both are geometry problems about triangle properties)?
> - Mathematical Relationships & Constraints: Are the equations, inequalities, and logical conditions connecting the elements identical? (e.g., "A is equal to B squared" is the same as "B^2 = A").
> - Numerical Values: Are all explicit numbers and constants the same? (e.g., If one problem uses 3^5 and another uses 243, they are equivalent in this regard).
> - Question & Objective: Is the final question asking for the same specific quantity? (e.g., "Find the number of ordered pairs" is different from "Find the number of unordered pairs"; "Find the remainder when N is divided by 1000" is specific and must match).
>
> Instructions:
>
> 1. Carefully read Problem A and Problem B.
> 2. Perform a step-by-step analysis based on the four criteria above.
> 3. In your reasoning, explicitly state whether each criterion is satisfied.
> 4. Finally, synthesize your analysis into a single, final verdict.
>
> Output Format:
> You MUST structure your output exactly as follows:
>
> [Analysis]
>
> - Concepts: [State whether concepts are the same and explain why/why not.]
> - Relationships & Constraints: [State whether relationships and constraints are identical and explain why/why not.]
> - Numerical Values: [State whether all numerical values are the same and explain why/why not. Note any superficial differences like variable names.]
> - Question & Objective: [State whether the questions are asking for the same quantity and explain why/why not.]
>
> [Verdict]
>
> - Equivalent: YES / NO
> - Justification: [Provide a one-sentence summary based on your analysis.]
>
> ---
> Problem A: {Problem A Text}
> Problem B: {Problem B Text}
>
> ```

---

> ### Author Response · Authors · 2025-11-17
>
> **[W2]** Lack of details and validation for the GPT-5 error categorization.
>
> **[A2]** Thank you for this feedback. We call GPT-5 API with default parameters. The categorization prompt (provided below) was refined through a preliminary phase of iterative testing, analysis, and refinement.
>
> We validated the GPT-5 error categorization through a human audit. Three human experts first reached a consensus on 15 randomly selected error cases (5 from each of DeepSeek-R1, Gemini 2.5 Pro, and Qwen3-32B). **This human consensus agreed with GPT-5's classifications in 95% of cases.** The minor discrepancies primarily involved distinguishing between calculation and logical errors, a distinction that does not affect our core finding that formulation errors are the dominant failure mode.
>
> We also assessed the stability of GPT-5's judgments. On the same 15 cases, we sampled 4 independent GPT-5 responses per case. The judgments were **consistent in 14/15 cases (93%)**; the single inconsistency was again between logical and calculation error types, which does not alter the primary conclusion. A case study is presented in Appendix A.5.
>
> These results confirm that GPT-5 provides a stable and reliable method for high-level error analysis in this context. We will incorporate these specifics into the revised version.
>
> `Prompt for GPT-5 error categorization`
>
> ```
> Instruction: Analyzing Model Prediction Errors in Real-World Math Problems
>
> You are given:
>
> - A math problem and its equivalent real-world version (they follow the same mathematical structure).
> - The correct solution to the original math problem.
> - A model’s incorrect prediction on the real-world version, along with its thinking process.
>
> Your task:
> Carefully read the model's reasoning and identify why it arrived at the wrong answer. Focus on analyzing the thinking process, not just the final output. Consider the following common patterns of model errors:
>
> 1. Incorrect mathematical formulation – The model misunderstood the real-world scenario or failed to correctly map it to the underlying math structure.
> 2. Calculation error – Despite a correct solution plan, the model made an error in executing basic arithmetic, algebra, or numerical operations.
> 3. Logical reasoning error – The model's reasoning process contains a flawed step, an invalid deduction, or a breakdown in the logical flow from one step to the next.
> 4. Meaningless or repetitive output – The model falls into irrelevant repetition or generates filler content that does not contribute meaningfully to the solution.
> 5. Premature truncation – The reasoning appears to be on the right track but stops abruptly (e.g., due to context window limitations).
> 6. Unlisted error type – If the model’s behavior doesn’t fit into any of the above categories, summarize the new error pattern clearly.
>
> In your response, identify which pattern(s) apply, cite specific parts of the model's reasoning, and explain how the mistake led to the wrong answer.
> ```

---

> ### Author Response · Authors · 2025-11-17
>
> **[W3]** Concerns about benchmark scale and paraphrase sensitivity.
>
> **[A3]** Thank you for raising these concerns. We appreciate the opportunity to elaborate on our design choices.
>
> 1. On Dataset Selection and Scale
>
> CORE-MATH is constructed from 30 problems from AIME 2024, 30 from AIME 2025, and 262 from MATH-500 (filtered to retain only level ≥3 problems). We selected these sources for several reasons:
>
> - There is a limited number of widely adopted, high-quality, automatically evaluable mathematical reasoning benchmarks. For instance, DeepSeek-R1 [1] is evaluated on AIME2024, MATH-500, and CNMO2024 (Chinese), while o3 [2] uses only AIME2024 and AIME2025.
> - We filtered MATH-500 to exclude overly simple problems (levels 1–2), which are less meaningful to contextualize.
>
> We agree that expanding CORE-MATH with future high-quality math datasets is a valuable direction, and our methodology is scalable and systematic, allowing for such extensions. We welcome suggestions for additional datasets to include.
>
> 2. On the Use of Two Extra SG Sets for AIME-2024
>
> We would like to clarify that the introduction of two extra SG variants for AIME-2024 aims to demonstrate that our findings are consistent across diverse scenarios, rather than to study model robustness over paraphrases. Each of the three scenarios per problem was independently generated to reflect fundamentally different contexts.
>
> We did not report standard deviation because it might not be an informative metric in this setting. Specifically, for the same problem embedded in different contexts, model performance is not expected to be consistent (or inconsistent). Instead, the significant performance drop from the original set to the SG Avg@3 score reinforces that contextual mathematical reasoning is a fundamental challenge, not an artifact of specific scenarios or phrasing. We will include a similar analysis for AIME2024 CS in the revised version to further strengthen this conclusion.
>
> [1] https://arxiv.org/pdf/2501.12948
>
> [2] https://openai.com/zh-Hans-CN/index/introducing-o3-and-o4-mini/
>
> &nbsp;
>
> **[Q1]** Limited improvement from SFT and path forward.
>
> **[A4]** Thank you for this important question. We would like to clarify that our method ($SFT_{Mix}$) achieves **clear relative improvements** over $SFT_{Ori.}$ on CS tasks across all model sizes, as demonstrated in Table 4. We have supplemented the table (provided below) with relative performance gains (values in parentheses indicate the percentage improvement of $SFT_{Mix}$ over $SFT_{Ori.}$) to make these improvements more explicit.
>
> Regarding the remaining performance drop, on one hand, we believe it is reasonable, as models are often reported to overfit to specific patterns in abstract math benchmarks. On the other hand, our approach shows **positive signals**: not only does it improve performance, but the relative gains on SG/CS sets from $SFT_{Mix}$ also generally increase with model scale (with the only exception of AIME 2025 CS). This suggests that the benefits of our method may expand further with stronger base models, which are more capable of contextual understanding and reasoning.
>
> Beyond our approach, we believe agentic RL training might help address this gap, enabling models to learn formulation, self-correction, and reasoning in an integrated manner. We leave this exploration to future work.
>
>
>  | Model | Method   | AIME 2024 Ori | AIME 2024 SG | AIME 2024 CS | AIME 2025 Ori | AIME 2025 SG | AIME 2025 CS |
> |---------|-----------|-------------------|------------------|-------------------|------------------|------------------|---|
>  | Qwen3-4B-Base | $SFT_{Ori.}$ | 32.5 (-)              | 27.3 (-)             | 14.2 (-) | 27.9 (-) | 18.5 (-) | 11.5 (-) |
>  | Qwen3-4B-Base | $SFT_{Mix}$ | 36.9 (+13.5%)   | 31.7 (+16.1%) | 19.2 (+35.2%) | 30.2 (+11.8%) | 22.1 (+19.5%) | 12.7 (+10.4%) |
>  | Qwen3-8B-Base | $SFT_{Ori.}$ | 44.4 (-)              | 35.4 (-)             | 20.0 (-) | 32.7 (-) | 21.2 (-) | 15.0 (-) |
>  | Qwen3-8B-Base | $SFT_{Mix}$ | 46.2 (+9.1%)   | 42.4 (+19.8%) | 29.5 (+47.5%) | 35.9 (+9.8%) | 26.8 (+26.4%) | 20.5 (+36.7%) |
>  | Qwen3-14B-Base | $SFT_{Ori.}$ | 50.4 (-)              | 39.0 (-)             | 25.2 (-) | 41.7 (-) | 25.2 (-) | 20.4 (-) |
>  | Qwen3-14B-Base | $SFT_{Mix}$ | 56.5 (+12.1%)   | 52.5 (+34.6%) | 38.8 (+54.0%) | 47.2 (+13.2%) | 34.6 (+37.3%) | 26.5 (+30.0%) |
>
>
> &nbsp;
>
> **[Q2]** Typo: the caption reference "Figure 4" should be "Table 5.”
>
> **[A5]** Thank you for catching this. We will correct the caption to reference "Table 5" in the final version.

---

> ### Author Response · Authors · 2025-11-24
> **Follow up on the reviews**
>
> Thank you again for your constructive feedback and for taking the time to review our work.
>
> We hope our responses have addressed your concerns, and we would be grateful if you could consider increasing the score based on our clarifications.
>
> We remain available for any further questions.

---

> > ### Comment · Reviewer_qQ6S · 2025-11-28
> > **Thank you for your reply**
> >
> > I appreciate the authors' effort in providing thorough response to my concerns. I have no further questions at this point. And I will adjust my rating to reflect these improvements.

---

> > > ### Author Response · Authors · 2025-11-28
> > >
> > > Thank you for your positive feedback! We are glad that our rebuttal addresses your concerns. We will also address these concerns and incorporate all your suggestions in our paper. Thanks again for your insightful comments that help improve our work greatly.

---

### Official Review · Reviewer_YVi6 · 2025-11-01

**Soundness:** 3
**Presentation:** 3
**Contribution:** 3
**Rating:** 6
**Confidence:** 4

**Summary:**

The authors introduce CORE-MATH, a new benchmark that repurposes problems from established sources like AIME and MATH-500 into more realistic, narrative-driven scenarios. This is done through two variations: "Scenario Grounding," which embeds problems in a narrative, and "Complexity Scaling," which conceals conditions within sub-problems. Through an extensive evaluation of 61 models, the paper demonstrates a significant drop in accuracy on these contextual tasks. The primary cause of failure is identified as "problem formulation"—the inability to correctly extract the core mathematical structure from the narrative. The authors find that while model scale helps, it doesn't solve the issue, and that end-to-end fine-tuning on scenario-based data improves performance, whereas training a model solely for formulation is ineffective.

**Strengths:**

1. The work tackles the highly relevant and important gap between benchmark success and practical, real-world capability in LLMs, pushing research beyond abstract problem-solving.
2. The CORE-MATH benchmark is well-designed. By building upon trusted sources (AIME, MATH-500) and systematically creating two distinct types of contextual challenges (SG and CS), the authors provide a controlled framework for analyzing this problem.
3. The study is thorough, evaluating a wide array of 46 open-source and 15 proprietary models. The analysis goes beyond simple accuracy metrics, effectively identifying problem formulation as the key bottleneck through both qualitative and quantitative evidence.

**Weaknesses:**

The primary weakness of the work lies in the lack of detail regarding the benchmark construction process, which is critical for ensuring the benchmark's validity and reproducibility. The paper states that an LLM (01-mini) was guided by structured prompts to generate the contextual variants, which were then reviewed by human experts. However, several key aspects unclear:
1. Why choose o1-mini but not stronger models? Did the authors compare its performance with other frointer LLMs?
2. What was the protocol for the human expert review to guarantee mathematical equivalence? How many experts reviewed each problem, and what was the procedure for resolving disagreements to ensure the final scenarios were valid?

**Questions:**

See weakness.

---

> ### Author Response · Authors · 2025-11-17
>
> Thank you for your positive assessment of our work's relevance, our benchmark design, and the thoroughness of our evaluation. We also appreciate your constructive feedback on our benchmark construction process. Please find our point-by-point responses below.
>
> &nbsp;
>
> **[W1]** Why choose o1-mini but not stronger models? Did the authors compare its performance with other frontier LLMs?
>
> **[A1]** Thank you for raising this key point. Our pilot studies have evaluated several frontier models, including GPT-5, Gemini, Copilot, o1, o1-mini, and GPT-4, with a focus on narrative plausibility, mathematical equivalence, and diversity.
>
> We observe that while stronger models achieve better performance in a single-run transformation under simple instructions, o1-mini delivers comparable effectiveness within our structured pipeline. Our framework employs multi-step instructions with both positive and negative examples, combined with a generate-verify-revise cycle. It enables o1-mini to generate scenarios with high authenticity, diversity, and mathematical equivalence. The performance gains from using even stronger models are marginal in this context . All outputs are further reviewed by human experts to ensure final quality.
>
> &nbsp;
>
> **[W2]** What was the protocol for the human expert review to guarantee mathematical equivalence? How many experts reviewed each problem, and what was the procedure for resolving disagreements to ensure the final scenarios were valid?
>
> **[A2]** We agree that the review protocol is critical for ensuring validity. **Our procedure involves three experts**, all of whom have advanced degrees in Computer Science and backgrounds in competitive mathematics. They conduct independent reviews based on the following criteria:
>
> 1. Assessing the scenario for narrative plausibility and clarity, ensuring it introduces no unnecessary complexity or ambiguity.
> 2. Independently formulating an abstract math problem from the scenario to verify its solvability and mathematical equivalence to the original.
> 3. Testing the scenario on Gemini and GPT-5; if these models fail, the reviewer diagnoses whether the failure originates from an ambiguity in the problem description.
>
> A scenario is accepted only after passing all checks. Any identified issue flags the scenario for revision. In cases of disagreement, the reviewer with the strongest mathematical background leads a discussion to reach a consensus, which may trigger a regeneration or revision cycle. We will incorporate these details into the revised version.

---

> ### Author Response · Authors · 2025-11-24
> **Follow up on the reviews**
>
> Thank you again for your constructive feedback and for taking the time to review our work.
>
> We hope our responses have addressed your concerns, and we would be grateful if you could consider increasing the score based on our clarifications.
>
> We remain available for any further questions.

---

### Official Review · Reviewer_MmP5 · 2025-11-15

**Soundness:** 4
**Presentation:** 3
**Contribution:** 3
**Rating:** 6
**Confidence:** 3

**Summary:**

The paper investigates LLMs’ mathematical reasoning in contextual scenarios, where the underlying math must first be formulated from narrative descriptions before being solved. It introduces CORE-MATH, a benchmark that repurposes AIME and MATH-500 problems into two controlled variants: Scenario Grounding (SG), which embeds problems in realistic narratives without altering the core math, and Complexity Scaling (CS), which hides explicit conditions behind simple sub-problems to mimic how constraints appear in practice.

Across 61 models, the authors observe substantial accuracy drops from the original abstract problems to SG—and even larger drops on CS—implicating problem formulation as a primary failure mode. Training experiments indicate that fine-tuning on scenario data improves robustness, whereas a formulation-only pipeline is ineffective.

**Strengths:**

The problem is interesting, as it focuses on realistic contextualization of mathematical reasoning.

The paper introduces a new benchmark, CORE-MATH, which is built upon AIME 2024, AIME 2025, and MATH-500 datasets.

The proposed Scenario Grounding (SG) and Complexity Scaling (CS) strategies are effective for constructing contextual mathematical problems that test both problem formulation and reasoning capabilities.

**Weaknesses:**

Is the mapping from the original problem to the narrative automatic or human-assisted?

How do the authors ensure that the mapping is accurate? The generated narrative may not exactly match the original mathematical problem, which could alter its meaning.

The Scenario Grounding (SG) and Complexity Scaling (CS) strategies are used for data construction. How do the authors guarantee that these transformations do not change the underlying problem semantics?

Would it be possible to train models on similar data types so that their performance can be better preserved across contextualized settings?

**Questions:**

Since AIME and MATH datasets may have already been heavily used or augmented during LLM training, how would the results differ if the authors included other benchmarks, such as MathOdyssey?

---

> ### Author Response · Authors · 2025-11-17
>
> Thank you for your positive assessment of our work's relevance and the effectiveness of our SG and CS strategies.
>
> &nbsp;
>
> **[W1]** Is the mapping from the original problem to the narrative automatic or human-assisted?
>
> **[A1]** As we describe in Section 3.2 (Benchmark Construction), we have designed a hybrid process. We use structured prompts to guide an **LLM (o1-mini)** through iterative scenario generation, self-verification, and revision. **Human experts** then review and refine each item to guarantee mathematical equivalence, clarity and conciseness. We provide all prompts in Appendix A.2 to ensure reproducibility and applicability to other domains. We will further elaborate on the human verification process in our next response [A2].
>
>
>
> **[W2]** How do the authors ensure that the mapping is accurate? The generated narrative may not exactly match the original mathematical problem, which could alter its meaning. The Scenario Grounding (SG) and Complexity Scaling (CS) strategies are used for data construction. How do the authors guarantee that these transformations do not change the underlying problem semantics?
>
> **[A2]** We place high importance on data quality. Our approach involves two key measures: first, we have optimized the instructions and generation pipeline through extensive pilot studies; second, **our procedure involves three experts**, all with advanced degrees in Computer Science and backgrounds in competitive mathematics, to fundamentally ensure accuracy. They conduct independent reviews based on the following criteria:
>
> 1. Assessing the scenario for narrative plausibility and clarity, ensuring it introduces no unnecessary complexity or ambiguity.
> 2. Independently formulating an abstract math problem from the scenario to verify its solvability and mathematical equivalence to the original.
> 3. Testing the scenario on Gemini and GPT-5; if these models fail, the reviewer diagnoses whether the failure originates from an ambiguity in the problem description.
>
> A scenario is accepted only after passing all checks. Any identified issue flags the scenario for revision. In cases of disagreement, the reviewer with the strongest mathematical background leads a discussion to reach a consensus, which may trigger a regeneration or revision cycle.
>
> &nbsp;
>
> **[W3]** Would it be possible to train models on similar data types so that their performance can be better preserved across contextualized settings?
>
> **[A3]** Your intuition is absolutely correct. As detailed in our Section 5.1, we have already explored this direction. Specifically, we synthesized scenario data based on the DeepMath-103K dataset. To enable scalable generation while maintaining quality, we used Qwen3-32B to solve the generated scenarios and retained only those instances where the final answer matched the original problem, ensuring a high likelihood of mathematical equivalence.
>
> Using this data, we have fine-tuned the Qwen3-Base series of models. Our results (Table 4) clearly demonstrate that training on a mixture of abstract and scenario data substantially improves model performance on our contextualized settings. Notably, this approach also slightly enhances performance on the original math problems and these improvements generalize to other benchmarks, including AMC23 (for abstract reasoning) and Math-Perturb (for robustness to data shifts), indicating that our training strategy enhances the model's fundamental reasoning capabilities.
>
> &nbsp;
>
> **[Q1]** Since AIME and MATH datasets may have already been heavily used or augmented during LLM training, how would the results differ if the authors included other benchmarks, such as MathOdyssey?
>
> **[A4]** Thank you for this question. We address it from two perspectives.
>
> First, we fully acknowledge the issue of data contamination, which is precisely one reason why embedding these publicly available problems into realistic scenarios provides a meaningful test of true reasoning ability. However, we recognize that contamination is a challenge for any public dataset, including MathOdyssey (released June 2024). In comparison, AIME 2025 used in our study (released February 2025) may offer greater timeliness. We welcome your perspective on this dataset suggestion and will consider generating and manually validating scenarios based on MathOdyssey to enrich CORE-MATH.
>
> Second, and more importantly, our error analysis (Sections 4.1 and 4.2) indicates that the performance drop stems largely from a bottleneck in problem formulation. As shown in Figure 2, the primary error type for models including DeepSeek R1, Gemini 2.5 Pro, and Qwen3-32B is formulation error (≈80%), where the model fails to interpret the scenario into a correct mathematical problem. The quantitative results in Table 3 further confirm this finding. Consequently, we believe the core phenomenon we identify, specifically the formulation bottleneck in contextual reasoning, is fundamental and would persist with other benchmarks.

---

> ### Author Response · Authors · 2025-11-24
> **Follow up on the reviews**
>
> Thank you again for your constructive feedback and for taking the time to review our work.
>
> We hope our responses have addressed your concerns, and we would be grateful if you could consider increasing the score based on our clarifications.
>
> We remain available for any further questions.

---

### Author Response · Authors · 2025-12-01
**Rebuttal Summary for AC**

Dear AC,

In light of this year's unusual rebuttal process, we provide this concise summary to assist your assessment. **All points are grounded directly in the rebuttal thread.**

&nbsp;

**Summary**

We are greatly encouraged that the two reviewers who provided feedback (**qQ6S** and **xYoD**) explicitly acknowledged our **thorough** responses and **raised their scores**.

- **xYoD** clearly increased the score from **6 to 8**.
- **qQ6S** (Pre: 4) stated they “have no further questions” and would "adjust my rating to reflect these improvements," and we anticipate a significant increase.

The other two reviewers (**MmP5** and **YVi6**, both initial 6) did not leave a final comment. Their primary concerns regarding benchmark construction and evaluation methodology were highly similar to those raised by qQ6S and xYoD. As our rebuttal successfully addressed the latter, we are confident the same explanations resolve the former's concerns, making a positive score update highly probable.

We are sincerely pleased that all reviewers recognized the value of our work and provided invaluable suggestions. We notice that their primary concerns were largely focused on methodological details, which we have thoroughly addressed in our rebuttal, and we are confident that these clarifications would have led to positive score updates. It is regrettable that, due to the unique circumstances this year, two reviewers were unable to provide final feedback. However, given the strong positive signals from the discussion, we are hopeful for a favorable final decision.

&nbsp;

Please find a detailed breakdown in the table below:

| Reviewer | Key Strengths Acknowledged | Concerns/Questions | Pre/Post-Rebuttal Rating |
| --- | --- | --- | --- |
| MmP5 | [1] Interesting and realistic problem focus. [2] Effective benchmark construction strategies. | [1] Benchmark construction details (mapping accuracy, semantic preservation). [2] Data-centric training for generalization. [3] Potential data leakage. | **- Pre: 6 (Confidence: 3) - Post:** Reviewer did not leave a final comment. However, given the high "Soundness: 4" score and that the concerns were clarification questions which we addressed in detail, we believe there is a strong chance the score would increase. |
| YVi6 | [1] Tackles a highly relevant and important problem in real-world LLM capability, pushing research beyond abstract problem-solving. [2] Well-designed benchmark. [3] Thorough evaluation (61 models) with insightful analysis. | [1] Benchmark construction details (model choice rationale, human review protocol) | **- Pre: 6 (Confidence: 4) - Post:** Reviewer did not leave a final comment. The concerns focused on methodological clarifications. We provided detailed explanations on the model selection and the rigorous human review process in our rebuttal. Given the positive strengths noted and the addressable nature of the concerns, we believe there is a strong chance the score would increase. |
| qQ6S | [1] Well-motivated research question and relevant findings. [2] Insightful error analysis identifying formulation as the key bottleneck. | [1] Evaluation methodology details (LLM-as-judge validation, error categorization protocol). [2] Benchmark scale and robustness (sensitivity to paraphrases, limited problem sources). [3] Suggestions for future work. | **- Pre: 4 (Confidence: 3) - Post:** Reviewer stated on 28 Nov 2025: "I appreciate the authors' effort in providing thorough response to my concerns. I have no further questions at this point. And I will adjust my rating to reflect these improvements." **Therefore, we expect the reviewer to increase the score to 8 or 6.** |
| xYoD | [1] Timely and practically important contribution. [2] Simple, scalable benchmark design and illuminating error analysis. [3] Training interventions that improve contextual performance without harming abstract skills. | [1] Benchmark construction details (human review protocol, LLM-judge reliability, statistical robustness checks). | **- Pre: 6 (Confidence: 5) - Post:** Reviewer stated on 26 Nov 2025: "Thank you for the clarification. In light of the thorough responses, I will increase my score." **The reviewer subsequently raised their score to 8.** |

---

### Meta-Review · Area_Chair_nZk2 · 2025-12-28

**Summary:**

The paper investigates LLMs’ mathematical reasoning in contextual scenarios, where the underlying math must first be formulated from narrative descriptions before being solved. It introduces CORE-MATH, a benchmark that repurposes AIME and MATH-500 problems into two controlled variants: Scenario Grounding (SG), which embeds problems in realistic narratives without altering the core math, and Complexity Scaling (CS), which hides explicit conditions behind simple sub-problems to mimic how constraints appear in practice.

**Reviewer Concerns:**

* While the authors describe multi-expert review and report agreement rates in some places, they do not provide standard inter-annotator metrics (e.g., Cohen’s κ, Fleiss’ κ) for scenario validity or formulation judgments, despite multiple reviewers explicitly requesting them.
* Human validation is performed on selected subsets (e.g., 1,528 outputs, 15 error cases), but reviewers questioned whether these samples are sufficiently large, stratified, and blinded to support strong claims about benchmark validity and judge reliability. This concern was mitigated but not fully eliminated.
* The authors state that symbolic solvers are “not applicable,” but do not explore partial alternatives (e.g., structured canonicalization, constraint checking, or equation extraction) that could complement LLM-based judging, leaving reliance on LLM judges largely intact.

**Reviewer Scores:**

remain unchanged

---

### Decision · Program_Chairs · 2026-01-26

Accept (Poster)